# SARL: Structure-Aligned Reinforcement Learning for Bridging the Perception-Action Gap in Airspace

**Binhao Gu** [1]   **Jinjun Cai** [1]   **Weihuang Zheng** [1]   **Jiaxing Li** [1]   **Youyong Kong** [1]   **Hui Ding** [2]

## Abstract

Multi-Agent Reinforcement Learning (MARL) has been widely applied to automated aircraft conflict resolution due to its strong capability for co-operative control and distributed decision-making. However, existing approaches typically assume a fixed number of aircraft and neglect the unique characteristics of air traffic control instructions. This structural misalignment between model architectures and domain requirements leads to severe deficiencies in perception scalability and action stability across scenarios of varying scales. To address these challenges, we propose Structure-Aligned Reinforcement Learning (SARL), which aims to bridge the gap between perception and action. First, the Physics-Encoded Relational Graph (PERG) effectively resolves the fixed input dimensionality issue by incorporating physical inductive biases into a graph attention mechanism. Second, we design the Sparse Cognitive Mixture-of-Experts (SC-MoE) to enhance decision stability. In addition, we introduce a Kinematic Safety Shield (KSS) based on aviation rules, which not only improves inference-time safety but also effectively guides the model to generate semantically meaningful actions that comply with aviation standards. Simulation experiment results demonstrate that SARL significantly outperforms existing reinforcement learning baselines across diverse scenarios in terms of both success rate and operational efficiency.

## 1. Introduction

Although modern Air Traffic Management (ATM) systems have achieved a high level of maturity in scheduling-based

---
[1]Southeast University [2]Nanjing Les Information Technology. Correspondence to: Youyong Kong <kongyouyong@seu.edu.cn>, Hui Ding <dhlshy2006@163.com>.

*Proceedings of the $43^{rd}$ International Conference on Machine Learning*, Seoul, South Korea. PMLR 306, 2026. Copyright 2026 by the author(s).

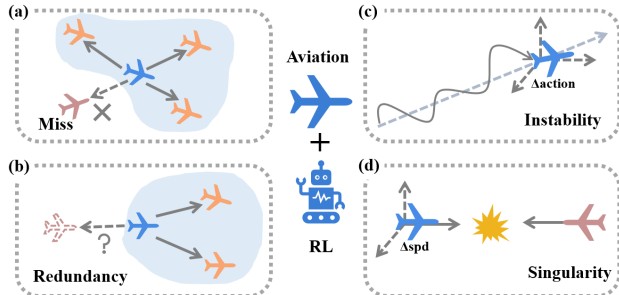

*Figure 1.* **Structural misalignment** between RL and the aviation field. (a) and (b) illustrate that fixed inputs can lead to missing nearby aircraft or input redundancy in different airspace, i.e., **perceptual misalignment**; (c) and (d) illustrate that coupled actions can lead to flight instability, and a single action cannot complete collision avoidance, i.e., **action misalignment.**

traffic optimization, such as the Traffic Management Advisor (TMA) and Time-Based Flow Management (TBFM) systems deployed in the United States, these approaches rely heavily on static rules and preplanned scheduling mechanisms, and therefore lack adaptability. This limitation becomes particularly pronounced with the introduction of advanced air traffic systems, which significantly increase airspace density and conflict frequency (Park et al., 2023). Developing efficient and scalable conflict resolution methodologies is therefore imperative to mitigate collision risks.

Reinforcement Learning (RL) offers a promising paradigm for ATM. In contrast to traditional scheduling-based schemes, RL enables the real-time acquisition of flight policies capable of adapting to dynamic and uncertain airspace environments. In single-encounter scenarios, this technology has demonstrated effectiveness in learning vector-based control policies for conflict resolution (Pham et al., 2019). At a broader control level, RL extends to Multi-Agent Reinforcement Learning (MARL), facilitating cooperative separation assurance among multiple aircraft in both structured (Brittain & Wei, 2022) and unstructured (Groot et al., 2025) airspace configurations. These approaches typically formulate the problem within a Markov Decision Process (MDP) framework, modeling aircraft as interacting agents (Brittain & Wei, 2019). The flight status of each aircraft, comprising position, airspeed, and heading, is encapsulated in state

vectors, providing critical information for agent decision-making. In this context, the fundamental objective of RL is to derive an optimal policy that maps aircraft states to specific conflict resolution maneuvers.

However, existing aviation RL paradigms suffer from significant **structural misalignment** when addressing aircraft conflict, primarily manifested as a **perception–action gap**, as illustrated in Figure 1. ***On the perception side***, current methodologies often overlook the dynamic nature of airspace, relying on the naive concatenation of state vectors into fixed-dimensional inputs (Yang & Wei, 2020; Brittain & Wei, 2021; Lai et al., 2021; Aziz & Wei, 2026). Given that the number of aircraft within a sector varies over time, such rigid concatenation severely constrains model scalability across diverse scenarios. Furthermore, at any given moment, conflicts may involve only a sparse subset of aircraft pairs. Consequently, the decision-making process is burdened by substantial data redundancy, raising critical concerns regarding learning efficiency and scalability. ***On the action side***, the majority of studies utilize only a single action dimension (Aziz & Wei, 2026; Li et al., 2024). Even approaches that incorporate multi-dimensional actions (Groot et al., 2025; Li et al., 2025) fail to systematically align these actions with established ATC protocols. This contradicts the inherent sparsity and decoupling of ATC instructions (Vela et al., 2012), where human controllers typically issue a discrete command along a single dimension (e.g., heading or speed) at any given time. Such semantic coupling within the action space not only precipitates an explosion in the action search space but also induces oscillatory trajectories. Finally, the decision-making processes of existing methods are predominantly black-box in nature (Guo & Wei, 2022). This lack of transparency severely impedes trust establishment, thereby restricting the deployment of RL algorithms in safety-critical scenarios, particularly those requiring human-in-the-loop interaction.

To bridge the perception–action gap, we propose a novel architecture termed **Structure-Aligned Reinforcement Learning (SARL)**. Inspired by Graph Neural Networks (GNNs), we introduce a **Physics-Encoded Relational Graph (PERG)** that models input states as a dynamic and computationally efficient graph structure. By incorporating a graph attention mechanism infused with physical inductive biases, the PERG captures evolving airspace information and effectively addresses perception scalability challenges. For action generation, we design a **Sparse Cognitive Mixture-of-Experts (SC-MoE)** action head. By leveraging a differentiable routing mechanism to dynamically select among decoupled expert modules, this architecture yields sparse control behaviors. To further enhance operational safety, we integrate a simplified **Kinematic Safety Shield (KSS)** based on selected airspace management regulations issued by authorities such as the International Civil Aviation

Organization (ICAO). The KSS is capable of inspecting and modifying model actions during inference, while also providing trigger signals to the reward function during training to encourage the learning of safer conflict resolution behaviors. Experimental results demonstrate that SARL exhibits outstanding scalability across scenarios of varying scales, significantly outperforming other reinforcement learning methods in terms of both success rates and efficiency.

The main contributions of this work are summarized as follows:

- We introduce a **physics-encoded relational graph** into the state encoder to align perceptions, using a graph attention mechanism to efficiently capture dynamic inter-aircraft interactions.

- To align actions, we develop a **Sparse Cognitive Mixture-of-Experts** module that dynamically activates decoupled action experts, yielding sparse and semantically structured control.

- We propose a **Kinematic Safety Shield** based on aviation collision-avoidance rules, which effectively guides the model to learn compliant actions during training while simultaneously enhancing safety during the inference process.

- Results from multiple simulation scenarios show that our **Structure-Aligned Reinforcement Learning** outperforms the baseline methods in terms of operational performance. Further analysis indicates that PERG effectively adapts to environments with varying traffic densities, while SC-MoE generates actions that exhibit sparsity and decoupling. These results verify that **SARL** effectively achieves structural alignment.

## 2. Related Works

**Traditional Conflict Resolution** Given the critical importance of aviation safety, substantial scholarly effort has been devoted to the problem of conflict resolution. As summarized in recent surveys (Kuchar & Yang, 2002; Jenie et al., 2016), existing methodologies can be broadly categorized into two streams. Mathematical Programming approaches formulate conflict resolution strategies as decision variables, constructing Mixed Integer Programming (MIP) models to minimize trajectory deviations (Pallottino et al., 2002; Pelegrín & d'Ambrosio, 2022). These formulations are typically solved using exact algorithms, such as Branch and Bound (Cafieri et al., 2023) and Column Generation (Hassan et al., 2021), or heuristic techniques, including Simulated Annealing (Courchelle et al., 2019) and Genetic Search (Wang et al., 2020). Although these methods can provide

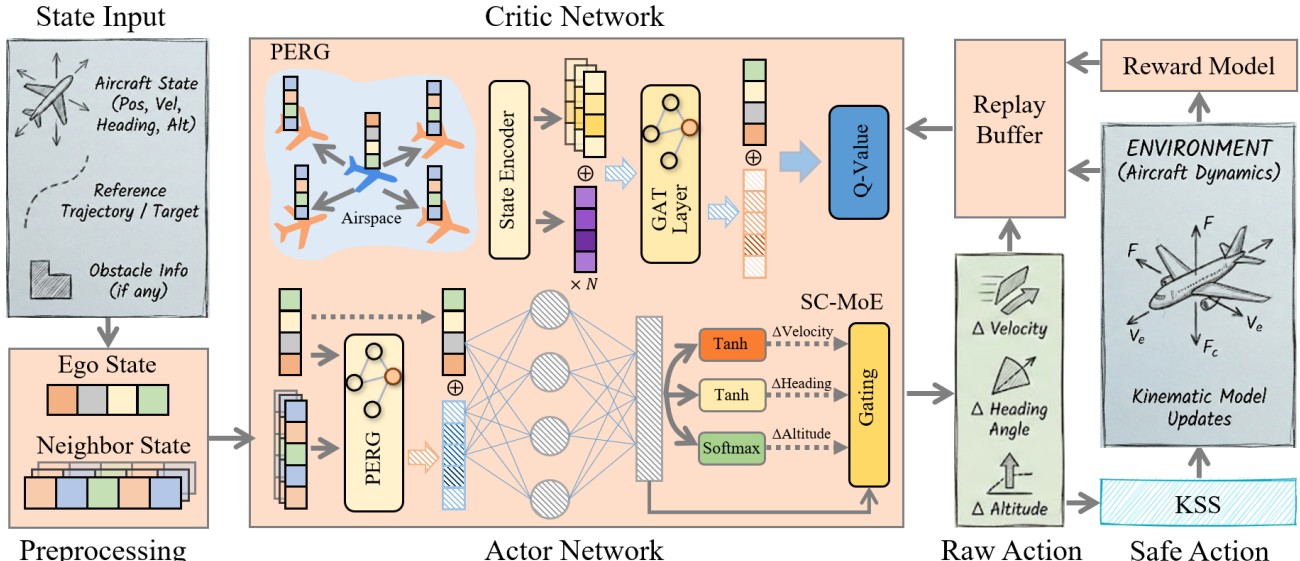

*Figure 2.* Overview of the proposed Structure-Aligned Reinforcement Learning (SARL).

theoretical optimality guarantees, their NP-hard computational complexity leads to prohibitively expensive solution procedures, rendering them unsuitable for the real-time requirements of high-density airspace. Conversely, Reactive Methods, including negotiation algorithms (Pritchett & Genton, 2017), geometric rules (Alharbi et al., 2020), and Artificial Potential Fields (Zeghal, 1998), have attracted significant attention due to their millisecond-level response times. However, as extensively noted in the literature (Ribeiro et al., 2020; Tang, 2019), these approaches lack a global perspective; consequently, they cannot guarantee the existence of collision-free solutions and are prone to converging to local optima or deadlocks in complex scenarios.

**Multi-Agent Reinforcement Learning** In the domain of airspace management, RL provides a powerful paradigm for addressing high-dimensional and complex decision-making problems (Ye et al., 2023; Selvaraj et al., 2022). Through continuous interaction with the environment, RL agents can distill decision-making knowledge and generalize learned policies to unseen scenarios (Ming et al., 2023; Chu et al., 2022). Despite the growing interest in applying RL to aviation conflict resolution (Wang et al., 2022; Tong et al., 2021), existing studies still suffer from severe structural misalignment. This misalignment manifests primarily in two aspects. From a perception perspective, both single-agent and multi-agent frameworks predominantly rely on naively concatenating the states of multiple aircraft into fixed-dimensional vectors (Lai et al., 2021; Aziz & Wei, 2026). Such an approach ignores the time-varying number of aircraft in the airspace; the resulting rigid input representations lack essential relational inductive biases, thereby severely limiting model efficiency and scalability across

scenarios of varying scales. From an action perspective, existing algorithms typically generate densely coupled control signals without regularization grounded in air traffic control logic (Li et al., 2024; Groot et al., 2025). This design contradicts the inherent sparsity and decoupling of aviation instructions (Vela et al., 2012). Although Mixture-of-Experts (MoE) architectures have demonstrated effective computational decoupling in natural language processing, their potential in aviation control remains largely unexplored. The SARL framework proposed in this work explicitly targets these forms of structural misalignment in both perception and action, systematically bridging the gaps through the integration of physics-encoded relational graphs and sparse cognitive control mechanisms.

## 3. Methodology

Figure 2 illustrates the overall architecture of the proposed SARL framework. SARL integrates PERG to improve the scalability of state representations and incorporates SC-MoE to enable sparse and decoupled action generation. In addition, KSS functions as a safety-critical module: during inference, it enforces hard constraint corrections via reachability analysis, while during training it encourages the policy to internalize safety boundaries, thereby enhancing collision avoidance.

Specifically, PERG is introduced as the perception encoder to address the limitation of fixed-size neighbor representations. By leveraging a graph attention mechanism to capture inter-aircraft interactions, PERG enables flexible aggregation of variable-length inputs, significantly improving scalability in large-scale airspace scenarios. In the

decision-making stage, SC-MoE generates the raw action $a_i^{\text{raw}}$ through multiple expert networks and a routing mechanism, achieving action decoupling and sparsification, which leads to smoother maneuvering behaviors.

To ensure compliance with air traffic control regulations, we further incorporate a rule-based KSS. Acting as a forward-looking safety filter, KSS corrects $a_i^{\text{raw}}$ to produce a safety-enhanced action $a_i^{\text{safe}}$. The correction signal is also fed back during training, guiding the agent toward safer and more aviation-aware policies.

### 3.1. Physics-Encoded Relational Graph

To mitigate the curse of dimensionality and the absence of topological inductive bias characteristic of naive state concatenation in traditional methodologies, we construct the PERG. Specifically, we model the local airspace as an ego-centric star-structured graph. In this topology, neighboring aircraft function as one-hop nodes, while edges denote potential interactive relationships. By leveraging a graph attention mechanism, the model dynamically computes edge weights to selectively aggregate critical neighborhood information.

**Feature Encoding and Embedding** To construct a high-fidelity representation of the airspace state, we employ differentiated feature embedding strategies for the ego aircraft and its neighbors. For the ego node $v_i$, the feature embedding $h_i$ is derived by directly encoding its intrinsic state $o_i^{ego}$ via a learnable linear projection layer. For neighbor nodes $v_j$, the feature construction comprises two distinct dimensions. First, we extract relative features—specifically to ensure translation invariance. Second, to inject domain knowledge, we explicitly incorporate Physics-Aware Flags. Specifically, we translate physical constraints derived from ATC regulations into binary feature signals, which are embedded within the node representations. This design introduces a physical inductive bias, significantly enhancing the network's sensitivity in identifying high-risk neighbors that violate safety boundaries. Formally, the feature computation is defined as follows:

$$h_j = \mathcal{H}\left(\$\left(o_i^{ego} \oplus o_i^j\right), P\right) \quad (1)$$

Here, $\$(\cdot)$ corresponds to relative feature extraction, $P$ denotes the Physics-Aware Flags, and $\mathcal{H}(\cdot)$ represents the resulting feature encoding for the neighbor nodes.

**Dynamic Relational Aggregation** To adaptively process variable-length neighbor sequences and extract critical information from complex airspace contexts, we employ a Graph Attention Mechanism (GAT). This approach is predicated on the core hypothesis that neighboring aircraft do not exert uniform influence on the ego-agent's decision-making. Consequently, we explicitly interpret the attention coefficient $\alpha_{ij}$ as a quantitative metric representing the potential conflict threat level posed by neighbor $j$ to the ego-agent $i$:

$$\alpha_{ij} = \frac{\exp\left(e_{ij}\right)}{\sum_{k \in N_i} \exp\left(e_{ik}\right)} \quad (2)$$

Here, $e_{ij}$ denotes the raw affinity score between the neighbor and the ego aircraft. It is computed by concatenating the ego feature embedding $h_i$ with the neighbor feature embedding $h_j$, and subsequently mapping the concatenated vector to a scalar via a learnable attention vector $\mathbf{a}$:

$$e_{ij} = \text{LeakyReLU}\left(\mathbf{a}^\top\left[\mathbf{W}_q h_i \parallel \mathbf{W}_k h_j\right]\right) \quad (3)$$

Here, $\mathbf{W}_q$ and $\mathbf{W}_k$ denote the learnable linear projection matrices that map the ego and neighbor features, respectively, into a shared latent space. The symbol $\parallel$ represents the vector concatenation operation, $\mathbf{a}$ is the attention weight vector, and LeakyReLU is employed to introduce non-linear activation.

The final context vector $c_i$ is computed as the weighted sum of neighbor features:

$$c_i = \sum_{j \in N_i} \alpha_{ij} h_j \quad (4)$$

This mechanism guarantees permutation invariance with respect to the input sequence, effectively empowering the model to adaptively accommodate airspace traffic of arbitrary density.

### 3.2. Sparse Cognitive Mixture-of-Experts

The SC-MoE module is designed to emulate the Cognitive-Executive dual-layer decision-making paradigm characteristic of human air traffic controllers. This mechanism proceeds from intention inference to precise control. The network architecture comprises a shared feature encoder, a differentiable Cognitive Router, and a set of functionally decoupled Expert modules.

**Shared Representation Encoding** To establish a unified state observation baseline, we first employ a Shared Feature Encoder to map the context features $c_i$ (aggregated by PERG) and the ego state $h_i$ into a high-dimensional latent space. The joint feature representation $z_i$ is computed as follows:

$$z_i = \text{ReLU}(\mathbf{W}_e[h_i \parallel c_i] + \mathbf{b}_e) \quad (5)$$

Here, $\mathbf{W}_e$ and $\mathbf{b}_e$ denote the learnable parameters of the encoder. This layer effectively integrates local dynamics with global topological information, thereby providing rich contextual semantics for the subsequent decoupled decision-making process.

**Heterogeneous Action Experts** Guided by aerodynamic characteristics, we construct a set of Heterogeneous Expert Modules, denoted as $E_k(\cdot)$, which decouple the high-

dimensional action space into independent physical dimensions. Each expert is parameterized by a lightweight Feed-Forward Network (FFN):

The Speed Expert ($E_{spd}$) and the Heading Expert ($E_{hdg}$) function as *Continuous Control Experts*, tasked with managing continuous dynamic adjustments. These experts map the latent feature $z_i$ to control signals via a non-linear projection, utilizing a Tanh activation function to strictly constrain the output within the physically feasible domain of $[-1, 1]$:

$$a_{spd} = \tanh(\text{FFN}_{spd}(z_i)) \tag{6}$$

The Heading Expert adopts the same formulation.

Recognizing that flight level adjustments typically exhibit discrete command characteristics (e.g., "Descend to FL300"), direct regression methods frequently induce control oscillation. Consequently, we formulate the Altitude Expert ($E_{alt}$) as a *Pseudo-Continuous Altitude Expert*. This module initially generates a ternary probability distribution $p_{alt} \in \mathbb{R}^3$, corresponding to the actions: Descend, Maintain, and Climb:

$$p_{alt} = \text{Softmax}(\text{FFN}_{alt}(z_i)) \tag{7}$$

Subsequently, this distribution is mapped back to the continuous space via a fixed reference projection vector $\mathbf{v}_{ref} = [-1, 0, 1]^\top$:

$$a_{alt} = \sum_{j=1}^{3} p_{alt}^{(j)} \cdot \mathbf{v}_{ref}^{(j)} \tag{8}$$

This Expected Projection mechanism effectively integrates the stability of discrete semantics with the differentiability of continuous gradients.

**Differentiable Cognitive Routing** The Cognitive Router serves as the "Tactical Decision Core". It generates a four-dimensional logits vector $l \in \mathbb{R}^4$, corresponding to four mutually exclusive tactical intentions $\mathcal{T} = \{\text{Hold}, \text{Speed}, \text{Alt}, \text{Hdg}\}$. To achieve hard selection compliant with ATC regulations during the inference phase, while simultaneously preserving end-to-end differentiability during training, we employ the Gumbel-Softmax mechanism in conjunction with the Straight-Through Estimator.

**Sparse Action Synthesis** The final action vector $a_i \in \mathbb{R}^3$ is synthesized via a weighted combination of the expert outputs, governed by the routing weights. To achieve semantic decoupling, we define a set of Action Mask Matrices $M_k \in \mathbb{R}^{3 \times 1}$, which map the scalar expert outputs to the corresponding dimensions of the action vector (e.g., $M_{spd} = [1, 0, 0]^\top$). The final synthesized action $a_i$ is given by:

$$a_i = \sum_{k \in \mathcal{T}} \omega_o^{(k)} M_k \cdot a_k(z_i) \tag{9}$$

Since $\omega_o$ is a One-Hot vector, only a single term in the equation remains non-zero at any given time step $t$. This implies

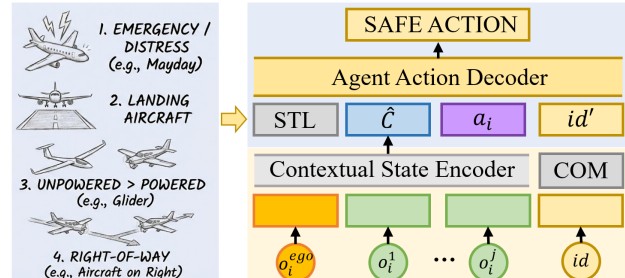

*Figure 3.* The overall architecture of the Kinematic Safety Shield.

that the agent strictly enforces Semantic Sparsity, characterized by a "Single-Moment, Single-Instruction" paradigm, while simultaneously eliminating control oscillation induced by multi-dimensional action coupling.

### 3.3. Kinematic Safety Shield

While the SC-MoE module delivers efficient policies, it does not fully guarantee that conflict resolution is executed in strict accordance with aviation regulations. To address this limitation, we introduce the rule-based KSS module, designed to enforce the normative compliance of the decision model's output actions while simultaneously enhancing overall operational safety. The structure of the KSS is illustrated in Figure 3.

**Rule Base Construction** The aviation domain is governed by explicit regulatory frameworks, such as the ICAO standards. To address this, we incorporate Signal Temporal Logic (STL) to construct a comprehensive rule base $\Phi$ that rigorously formalizes both the safety regulations and the right-of-way rules.

**Contextual State Encoder** During flight operations, aircraft typically lack access to the global context and the complete distribution of other agents. Consequently, it is necessary to construct a unified model of the observable environment based on local observations to align with Rule Base $\Phi$. Specifically, the input to the Contextual State Encoder $f^{Encoder}$ comprises the ego-agent's self-observation $o_i^{ego}$ and the state information of neighboring aircraft $o_i^{nig}$:

$$\hat{c} = f^{Encoder}\left(o_i^{ego}, o_i^{nig}\right) \tag{10}$$

This contextual representation does not aim to directly reconstruct the complete ground-truth state $c$. Rather, it achieves two primary objectives: (1) mapping the complex state space into a compact low-dimensional representation; and (2) aligning the environmental context with the Rule Component Library.

**Agent Action Decoder** Upon generating the contextual information, the subsequent critical phase focuses on assigning a unique identity identifier to each agent. Most existing

MARL frameworks rely on heuristic rules to determine agent identity.

However, in scenarios characterized by contextually homogeneous features, such methods often struggle to guarantee identity heterogeneity, potentially resulting in the generation of identical actions. To address this, leveraging the unique ID of the aircraft, we first utilize a custom-designed Comparator Com to generate a transformed identifier $id'$. This identifier is subsequently integrated as a reference feature into our Agent Identity Decoder $f^{Decoder}$. Simultaneously, by synthesizing the decoder output $\hat{c}$ and the Rule Base $\Phi$, the module encodes the raw action $a_i$ from SC-MoE into a safety-compliant action $a_i^{safe}$ that strictly adheres to aviation regulations:

$$a_i^{safe} = f^{Decoder}(\hat{c}||\Phi||a_i||Com(id)) \quad (11)$$

**Learning Objective** To prevent the agent from developing an over-reliance on the Shield, we treat the intervention of the KSS as a soft failure. Accordingly, the reward function component $R_s$ is designed as follows:

$$R_s = -\mathbb{I}_s \cdot C_{intervene} \quad (12)$$

Here, $\mathbb{I}_s$ serves as the Shield activation indicator variable, and $C_{intervene}$ represents a negative penalty term. This formulation incentivizes the agent to proactively learn and adopt the safety-compliant actions guided by the KSS.

# 4. Experiments

This section presents a systematic evaluation of the SARL within the high-fidelity air traffic control simulation platform, BlueSky (Hoekstra & Ellerbroek, 2016). To comprehensively validate the model's effectiveness across perception, decision-making, and safety, we designed an experimental protocol comprising baseline comparisons and ablation studies, aimed at addressing the following four core Research Questions (RQs):

- **RQ1 (Operational Performance):** Compared to other methods, does SARL demonstrate superior performance in air traffic conflict resolution through structural alignment and safe guidance?

- **RQ2 (Perception Scalability):** Can the PERG module overcome the limitations of state concatenation and demonstrate superior scalability across dynamic airspaces of varying scales?

- **RQ3 (Action Compliance):** Can the SC-MoE mechanism effectively decouple control dimensions to generate compliant actions that adhere to air traffic control logic (characterized by sparsity and smoothness)?

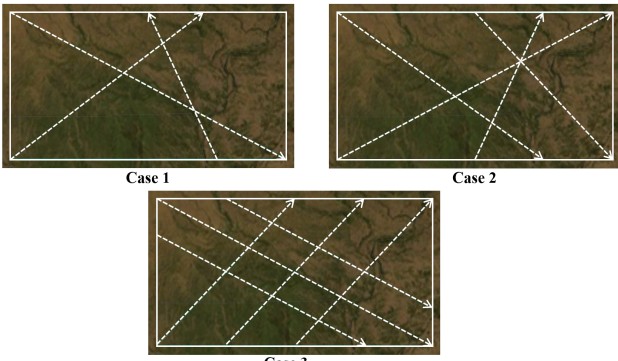

*Figure 4.* Example of the experimental scenario

- **RQ4 (Safety Assurance):** Does the introduction of the KSS module significantly reduce collision rates without sacrificing operational efficiency?

## 4.1. Implementation Details

**Evaluation Metrics** In the experiments, we define the following metrics to comprehensively assess model performance, efficiency, and robustness. (1) **Success Rate** (*Succ Rate*): the proportion of aircraft that successfully reach their destinations without collisions or exceeding the simulation boundaries. (2) **Average Operational Cost** (*Avg Op. Cost*): the weighted sum of control action magnitudes, used to evaluate control efficiency. (3) **Average Minimum Separation** (*Avg Min Sep*): the average of the global minimum pairwise distances among all aircraft. A higher value indicates that the model tends to proactively resolve conflicts at larger separation distances, reflecting a more robust policy.

**Scenario Settings** To validate our model, we planned a $400km \times 200km$ rectangular airspace in the BlueSky simulation environment and constructed three sets of line segments with different numbers of routes and starting and ending points within the airspace, as shown in Figure 4. In the path planning process, we fully considered the three scenarios of aircraft conflicts mentioned in Appendix B.2.

To ensure the reliability and fairness of the results, all experiments are repeated with five different random seeds. In addition, task-specific hyperparameter tuning is conducted for all baseline methods to adapt them to the aircraft conflict resolution setting.

## 4.2. Operational Performance

To verify the performance of SARL in air traffic conflict resolution, we compared it with two types of baseline methods: (1) general-purpose MARL algorithms, including MAD-DPG (Lowe et al., 2017), DGN (Jiang et al., 2018), MATD3 (Fujimoto et al., 2018), and MAPPO (Yu et al., 2022); and (2) state-of-the-art aviation-specific methods, including SA

*Table 1.* Comparative experiments between **SARL** and other baseline methods across different cases. We mark the best result in bold and underline the second-best result.

| Env | Case1 | | | Case2 | | | Case3 | | |
|---|---|---|---|---|---|---|---|---|---|
| **Method** | *Succ Rate* | *Avg Op. Cost* | *Avg Min Sep* | *Succ Rate* | *Avg Op. Cost* | *Avg Min Sep* | *Succ Rate* | *Avg Op. Cost* | *Avg Min Sep* |
| **MADDPG** | $94.84_{\pm1.32}$ | $345.56_{\pm5.09}$ | $17.32_{\pm0.07}$ | $93.74_{\pm0.22}$ | $370.41_{\pm9.22}$ | $\underline{18.60}_{\pm0.41}$ | $93.62_{\pm1.24}$ | $345.05_{\pm9.21}$ | $\underline{17.34}_{\pm0.36}$ |
| **DGN** | $89.86_{\pm1.30}$ | $235.23_{\pm27.65}$ | $16.31_{\pm0.22}$ | $91.06_{\pm0.32}$ | $\underline{265.52}_{\pm2.54}$ | $16.66_{\pm0.55}$ | $94.22_{\pm2.20}$ | $\underline{258.65}_{\pm12.63}$ | $15.08_{\pm0.63}$ |
| **MATD3** | $95.16_{\pm2.20}$ | $\underline{322.35}_{\pm12.21}$ | $19.14_{\pm2.18}$ | $92.32_{\pm1.90}$ | $295.03_{\pm12.29}$ | $16.53_{\pm0.31}$ | $\underline{95.10}_{\pm0.54}$ | $319.88_{\pm2.84}$ | $16.62_{\pm0.36}$ |
| **MAPPO** | $86.12_{\pm0.62}$ | $512.15_{\pm7.95}$ | $11.98_{\pm0.41}$ | $87.98_{\pm0.38}$ | $499.63_{\pm0.36}$ | $12.98_{\pm0.02}$ | $90.64_{\pm0.32}$ | $468.41_{\pm17.30}$ | $14.32_{\pm0.45}$ |
| **SA** | $92.40_{\pm0.14}$ | $351.22_{\pm33.59}$ | $14.81_{\pm0.64}$ | $92.44_{\pm0.30}$ | $351.72_{\pm30.58}$ | $14.88_{\pm0.53}$ | $92.70_{\pm0.72}$ | $351.04_{\pm14.92}$ | $17.09_{\pm0.19}$ |
| **LSRL** | $94.12_{\pm0.76}$ | $389.56_{\pm1.81}$ | $\underline{20.59}_{\pm0.61}$ | $92.64_{\pm0.36}$ | $343.13_{\pm6.90}$ | $17.69_{\pm1.17}$ | $92.82_{\pm2.08}$ | $347.07_{\pm11.27}$ | $14.87_{\pm0.34}$ |
| **SARL** | $\mathbf{99.28}_{\pm0.12}$ | $\mathbf{157.54}_{\pm4.24}$ | $\mathbf{24.39}_{\pm0.32}$ | $\mathbf{99.06}_{\pm0.80}$ | $\mathbf{100.36}_{\pm1.56}$ | $\mathbf{22.39}_{\pm0.45}$ | $\mathbf{97.00}_{\pm0.92}$ | $\mathbf{159.96}_{\pm19.90}$ | $\mathbf{24.80}_{\pm0.38}$ |

([Nilsson et al., 2025](#)) and LSRL ([Choi et al., 2025](#)). As shown in Table 1, SARL consistently outperforms all comparison methods in terms of success rate. Notably, while achieving the highest success rate, SARL also achieves the lowest operational cost. This indicates that SARL's safety is not achieved through aggressive maneuvering; instead, SARL utilizes MoE-based alignment and the KSS mechanism to learn a sparse optimal control strategy that adapts to aviation regulations, ensuring both improved safety and stable flight (supporting evidence is provided in Sections 4.4 and 4.5). Furthermore, the superior statistical data regarding average separation distance also demonstrates its strong safety robustness. SARL tends to make small, proactive adjustments at longer distances rather than performing sudden avoidance maneuvers at close range. This proactive decision-making behavior aligns closely with real-world air traffic control safety principles.

### 4.3. Perception Scalability

To empirically validate the superiority of the PERG module in dynamic airspace environments, we employ a fixed-size concatenation scheme for input processing in the baseline model. Conversely, we added PERG to the baseline to verify the effectiveness of its perceptual alignment. We construct three testing scenarios with different traffic densities, **Sparse**, **Medium**, and **Dense**. Environmental pressure is regulated by strictly controlling airspace capacity (with maximum numbers of aircraft set to 5, 10, and 18, respectively) and flight generation frequency (with inter-arrival intervals of 30, 20, and 10 simulation time steps). For the Baseline model, we conducted separate performance evaluations across each scenario using varying observation window sizes (fixed neighbor count $k \in \{1, \dots, 5\}$). In contrast, the model with PERG was evaluated on its ability to inherently process variable-length inputs without fixed window constraints.

As illustrated in Figure 5(a), the experiments reveal an intuitive phenomenon characterized by Information Saturation and Dimensional Mismatch. In the Baseline model, the success rate exhibits a trend of initial growth followed by

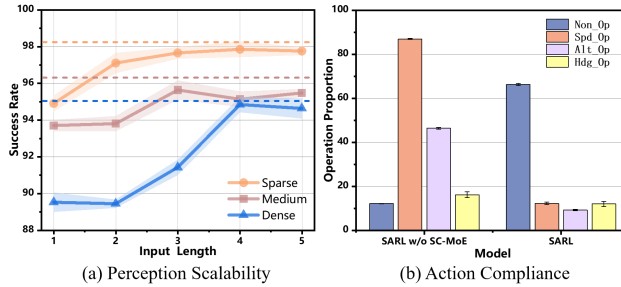

(a) Perception Scalability     (b) Action Compliance

*Figure 5.* SARL Experiments on Perception and Decision Alignment. *(a)* Results for perception alignment. The solid line represents the baseline, and the dashed line represents the results after adding PERG. *(b)* Results for action alignment.

saturation as the number of input neighbors $k$ increases. This indicates that the demand for information quantity varies dynamically across airspaces of different densities: low-density scenarios require minimal neighbor information, whereas high-density scenarios demand a broader field of view. Consequently, fixed-dimensional inputs lead to computational redundancy in sparse settings and Information Loss in dense settings. In contrast, the model with PERG added demonstrated superior adaptability. Leveraging the permutation invariance and attention mechanism of the PERG, it adaptively aggregates key neighbor features. The results indicate that the model with PERG consistently outperforms the performance of the Baseline, even when the latter is fine-tuned to its optimal window size, across all density scenarios. This provides compelling evidence that the PERG successfully bridges the structural misalignment issue in perception, achieving robust scalability across scenarios of varying scales.

### 4.4. Action Compliance

To validate alignment in action generation, we conduct an ablation study on the SC-MoE module. To isolate its effect, the KSS component is removed from SARL in all experiments. We conducted a statistical analysis of the model's action distribution and activation frequency on the test set, as illustrated in Figure 5(b).

The results show that SARL w/o SC-MoE exhibits typical characteristics of Action Coupling: its hold operations (Non-Op) account for a negligible proportion, whereas the aggregate activation rate of various adjustment actions exceeds 150% (implying that, on average, more than 1.5 action dimensions are activated simultaneously at each time step). This strategy, characterized by mixed micro-adjustments, not only induces trajectory oscillation but also severely violates ATC operational standards, consuming unnecessary communication bandwidth. Conversely, SARL w/ SC-MoE presents a radically different distribution. Following the introduction of SC-MoE, the proportion of Non-Op actions rises significantly, and the activation distribution of mutually exclusive actions exhibits high Semantic Sparsity. This confirms that SC-MoE effectively achieves the structural alignment between control outputs and aviation instructions.

### 4.5. Safety Assurance

To examine whether KSS can further improve safety without compromising the action alignment achieved by SC-MoE, we conduct an ablation study in case 1, comparing SARL with SARL w/o KSS. We primarily evaluate three metrics: success rate, collision rate, and average operational cost.

*Table 2.* Ablation Study of KSS. *Coll Rate* represents the probability of an aircraft collision.

| Method | Succ Rate | Coll Rate | Avg Op. Cost |
|--------|-----------|-----------|--------------|
| w/o KSS | 93.24±0.14 | 4.76±0.04 | 123.83±4.48 |
| SARL | 99.28±0.12 | 0.04±0.04 | 157.54±4.24 |

Table 2 shows that the inclusion of KSS significantly improves the task success rate of SARL while maintaining low operational costs (i.e., without sacrificing action sparsity). This indicates that KSS effectively guides the aircraft in conflict resolution through STL. To further verify the crucial strategic guidance role of KSS in conflict resolution: it not only actively intercepts dangerous actions but also guides the agent to learn expert-level air collision avoidance rules through reward shaping. We visualize the decision-making process in detail through the inherent interpretability of SARL, and the results are shown in Figure 6. Figure 6(a) shows a conflict-free scenario. SARL detects that the two adjacent aircraft are at a safe distance and will not collide, and that its own aircraft is strictly aligned with the flight path, so it chooses to accelerate to improve flight efficiency. Figure 6(b) shows a conflict resolution scenario. It displays the visualization results of different aircraft at the same moment. The two aircraft are at a critical point of near-collision, and they are able to consistently choose opposite actions to avoid collision.

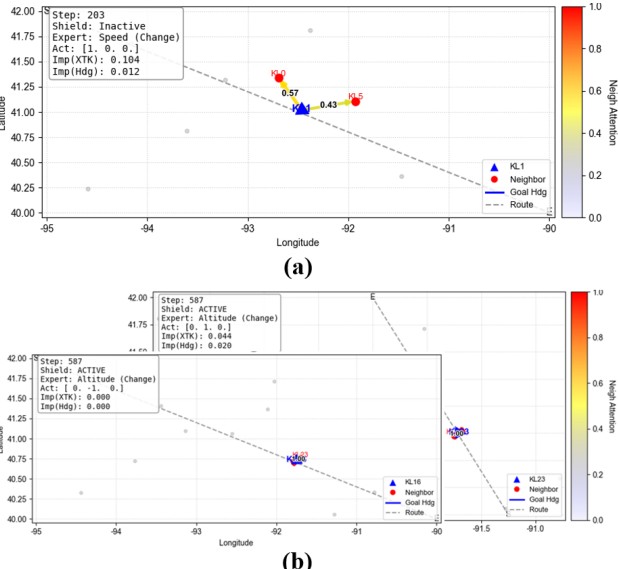

**(a)**

**(b)**

*Figure 6.* Aircraft flight visualization. *(a)* Shows that the aircraft detects no collision risk with adjacent aircraft and chooses to accelerate to improve efficiency; *(b)* Two aircraft detect each other within the collision range and adjust their altitudes upwards and downwards respectively to resolve the conflict.

## 5. Conclusion

This paper proposes SARL, a structured multi-agent reinforcement learning framework specifically designed to bridge the perception–action gap that arises when RL is directly applied to the aviation domain. Unlike conventional approaches that often overlook domain-specific characteristics, SARL emphasizes structural alignment between the model architecture and aviation domain properties. SARL effectively addresses the challenge of varying input dimensions through the graph attention mechanism in the PERG module. Meanwhile, the routing mechanism of the SC-MoE module enables task-specific action expert selection, achieving sparse semantic control that complies with aviation regulations. Building upon the structural alignment established by the first two modules, SARL further introduces the KSS module to enhance the safety of aircraft conflict resolution. The KSS module performs rule-based modifications on the model's actions and feeds rule-trigger events back to the learning process as reward signals. Experimental results demonstrate that SARL not only achieves superior safety and operational efficiency across multiple scenarios, but more importantly, exhibits inherent interpretability and regulatory compliance. SARL establishes a new technological paradigm for the development of trustworthy, safe, and human–machine collaborative air traffic control systems in the future. Our code is available at https://github.com/Gu-2002/SARL.

## Acknowledgements

This work is supported by 2024YFB4303805 National Key Research and Development Program of China. We would like to thank the reviewers for their constructive and insightful suggestions.

## Impact Statement

This paper presents work whose goal is to advance the field of Machine Learning. There are many potential societal consequences of our work, none which we feel must be specifically highlighted here.

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

# A. Implementation Details

## A.1. Algorithmic Description

The pseudo-code of SARL is shown in Algorithm 1.

---

**Algorithm 1** Structure-Aligned Reinforcement Learning (SARL) Training Procedure

---

**Input:** Scenarios $\mathcal{E}$, Max Episodes $K$, Batch Size $B$, Soft Update Rate $\rho$
**Output:** Trained Policy $\pi_\theta$

1   **Initialize:** PERG parameters $\phi$, SC-MoE Actor $\theta$, Critic $\psi$, Target Networks $\theta'$, $\psi'$, Replay Buffer $\mathcal{D}$
2   **for** *episode* $k = 1$ **to** $K$ **do**
3      Reset environment, observe states $\mathcal{S} = \{s_1, \ldots, s_N\}$ **for** *step* $t = 0$ **to** $T$ **do**
4         Construct ego-centric graph $\mathcal{G}_i^t$ based on detection range $R_{det}$ Extract physics-encoded representation $z_i \leftarrow$ PERG$_\phi(s_i, \mathcal{G}_i^t)$
5         Compute router logits $l \in \mathbb{R}^4$ and expert outputs $\{E_k(z_i)\}$ Sample noise $g \sim \text{Gumbel}(0, 1)$ Compute soft weights: $\omega_{soft} \leftarrow \text{Softmax}((l + g)/\tau)$ Compute hard weights: $\omega_{hard} \leftarrow \text{OneHot}(\text{argmax}(l + g))$ Apply STE: $\omega_o \leftarrow \omega_{hard} - \text{sg}(\omega_{soft}) + \omega_{soft}$ Synthesize sparse action $a_i^{raw} \leftarrow \sum \omega_o^{(k)} M_k E_k(z_i)$
6         **if** $KSS(s_i, a_i^{raw})$ *predicts imminent collision* **then**
7             $a_i \leftarrow a_{safe}$ $r_t \leftarrow r_t + C_{shield}$
8         **else**
9             $a_i \leftarrow a_i^{raw}$
10         **end**
11         Execute actions $a$, observe reward $r$ and next state $s'$ Store transition $(s, a, r, s')$ in $\mathcal{D}$
12      **end**
13      **if** $k \mod N_{update} == 0$ **then**
14         Sample mini-batch $B$ from $\mathcal{D}$ Update Critic $\psi$ by minimizing Bellman MSE Loss Update Actor $\theta, \phi$ via Policy Gradient (gradients flow through STE) $\theta' \leftarrow \rho\theta + (1 - \rho)\theta'$, $\psi' \leftarrow \rho\psi + (1 - \rho)\psi'$
15      **end**
16   **end**

---

## A.2. Hyperparameters

Unless specified otherwise, the hyperparameter configurations across different environments are presented in Table 3.

*Table 3.* Hyperparameter Settings for SARL and Baselines

| Category | Hyperparameter | Value |
|---|---|---|
| **Optimization** | Actor Learning Rate ($\alpha_\pi$) | $1 \times 10^{-4}$ |
| | Critic Learning Rate ($\alpha_Q$) | $3 \times 10^{-4}$ |
| | Optimizer | Adam |
| | Discount Factor ($\gamma$) | 0.99 |
| | Batch Size | 256 |
| | Replay Buffer Size | $5 \times 10^4$ |
| | Soft Update Rate ($\tau_{target}$) | 0.005 |
| | Training Episodes ($K$) | 50,000 |
| **Network (SARL)** | Hidden Dimension | 128 |
| | PERG Attention Heads | 4 |
| | PERG LeakyReLU Slope | 0.2 |
| | MoE Gumbel Temperature ($\tau$) | $1.0 \rightarrow 0.1$ (Annealed) |
| | Number of Experts | 3 |
| | Action Mask ($M_k$) | Fixed Identity |
| | Activation Function | ReLU / Tanh |

Continue on the next page

*Table 3.* Hyperparameter Settings for SARL and Baselines (2)

| Category | Hyperparameter | Value |
|---|---|---|
| **Environment & Safety** | Safety Distance ($D_{safe}$) | 5.0 KM |
| | Collision Penalty ($C_{collision}$) | -50.0 |
| | Shield Penalty ($C_{shield}$) | -10.0 |
| | Detection Range ($R_{det}$) | 20.0 KM |
| | Curriculum Threshold ($\delta_{sr}$) | 0.90 |

## B. Environment Details

### B.1. BlueSky

BlueSky is an open-source air traffic management simulator designed to improve the comparability and reproducibility of ATM research through open data and open-source code. It uses Python as its core programming language, supports multi-platform operation, and provides a concise text-based scenario definition language (TrafScript) and a user-friendly graphical interface, making it easy for users without computer science backgrounds to use and modify, as shown in Figure 7. The project integrates publicly available navigation data, is compatible with the BADA 3 performance model, and combines conceptual design with machine learning methods to build flight performance data, ensuring high fidelity while avoiding reliance on commercial data. BlueSky employs a modular design and vectorized computation, supporting large-scale real-time or fast simulations, and aims to provide a freely usable and collaboratively extensible simulation platform for academic and industry researchers.

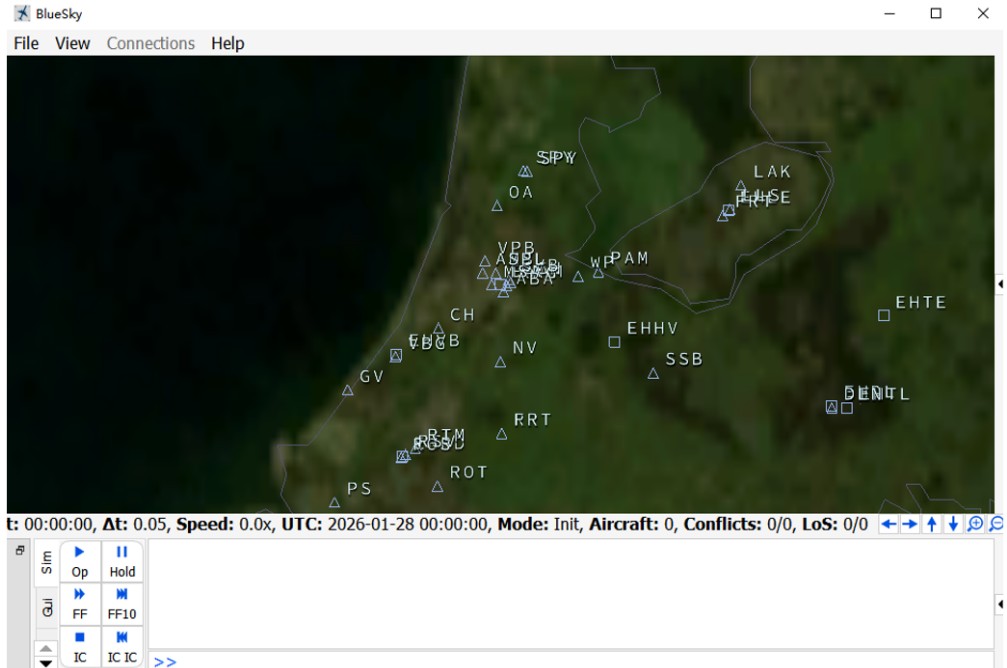

*Figure 7.* BlueSky Visual Interactive Interface.

Our code primarily involves the following commands:

**CRE acid, type, lat, lon, hdg, alt, spd**. Create an aircraft at specified coordinates.

**DEL acid/WIND/shape**. The DEL command is used to delete objects while running Bluesky.

**HDG acid,hdg (deg,True)**. Heading command (autopilot).

**ALT acid, alt, [vspd]**. Altitude command for the autopilot. Possibly also set the autopilot vertical speed.

**SPD acid,spd**. Speed command (autopilot) [CAS-kts/Mach]

## B.2. Scene Construction

In the aircraft scenario design, we primarily base our design on the three types of aircraft conflicts shown in Figure 8. Our scenarios aim to include all three situations simultaneously:

**Same-direction trajectory conflict:** Two aircraft are flying in similar directions, and due to speed differences or altitude changes, the trailing aircraft may gradually approach the leading aircraft. This scenario mainly tests the decision-making model's ability to maintain safe longitudinal separation and dynamically adjust speed and altitude strategies.

**Opposite-direction trajectory conflict:** Two aircraft are flying in opposite or nearly opposite directions, posing a risk of head-on collision. This type of conflict has the highest risk level and provides the shortest reaction time for air traffic controllers or intelligent decision-making systems. Therefore, it is a crucial scenario for verifying the model's ability to generate avoidance instructions under high-risk and time-critical conditions.

**Crossing trajectory conflict:** The flight paths of two aircraft intersect at a certain spatial point or region. If the timing is not properly coordinated, a conflict is highly likely. This scenario focuses on testing the model's comprehensive judgment and prediction capabilities regarding multi-dimensional spatio-temporal information, and how to select the optimal avoidance solution to minimize flight delays and extra fuel consumption while ensuring operational safety.

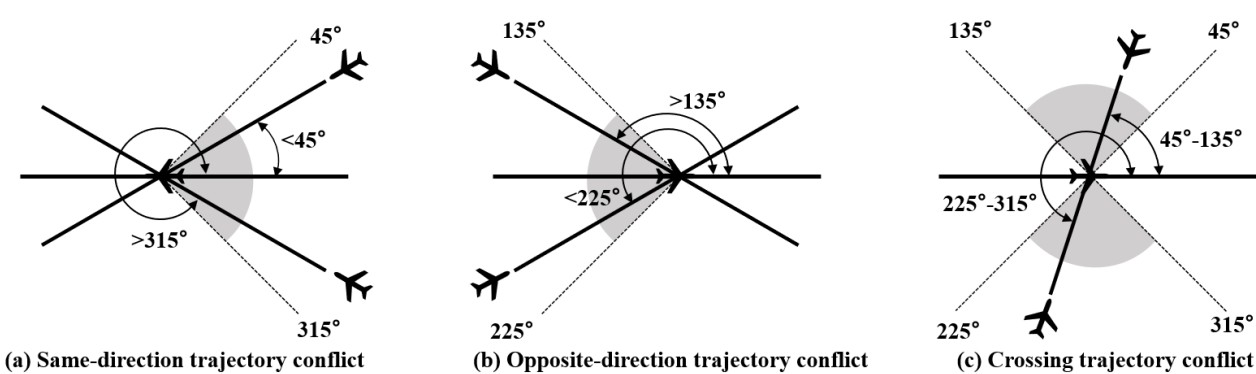

**(a) Same-direction trajectory conflict**   **(b) Opposite-direction trajectory conflict**   **(c) Crossing trajectory conflict**

*Figure 8.* Three aircraft conflict scenarios.

## C. Reinforcement Learning in Aviation

### C.1. Preliminaries

We formulate the multi-aircraft conflict resolution problem as a Decentralized Partially Observable Markov Decision Process (Dec-POMDP), defined by the tuple $\langle N, S, A, P, R, \Omega, O, n, \gamma \rangle$. Here, $N = 1, \ldots, n$ denotes the dynamically varying set of aircraft within the airspace. $s \in S$ represents the true global state of the environment, from which each aircraft agent extracts an individual observation $O(s, i)$ via the observation function $o_i \in \Omega$. At each time step, each agent $i$ selects an action $a_i \in A_i$ conditioned on its action-observation history $\tau_i \in T = (\Omega \times A)^*$, forming the joint action $a \in A^n$. This induces a transition to the subsequent state $s'$ according to the state transition function $P(s' \mid s, a) : S \times A \to \delta(S)$. All agents operate under a shared reward model $r = R(s, a)$, where $R : S \times A \to \mathbb{R}$ denotes the reward function and $\gamma \in [0, 1]$ represents the discount factor.

In the context of aircraft conflict resolution, agents operate under partial observability. Each agent perceives its own state $o_i^{ego} = \langle lat, lon, cas \ldots \rangle$ as well as a set of neighboring aircraft $o_i^{nig} = \{o_i^1, \ldots, o_i^k\}$ within a specified communication range, where $k$ denotes the cardinality of the observed neighbor set. The state of these neighbors is formalized as a tuple $o_i^j = \{lat_i^j, lon_i^j, \ldots\}$. In contrast to traditional methodologies that rely on densely coupled continuous vector outputs—simultaneously adjusting speed, heading, and altitude—we formulate the action space as a semantically sparse structure. Specifically, at any given time step, an agent executes only a single type of maneuver (or maintains its current state), thereby mimicking the single-instruction operational logic characteristic of human air traffic controllers. Regarding system dynamics, environmental state transitions inherently follow physical kinematic laws $\mathcal{T}$. The reward function $R$

incorporates components for trajectory tracking and collision penalties, along with a specialized term designed to enforce aviation collision-avoidance regulations.

## C.2. Reward Model

Here, we mainly introduce the design of reward functions $R$ in reinforcement learning. In our model, the reward function $R$ is designed to guide the agent towards conflict-free trajectory tracking while maintaining flight stability and adhering to kinematic constraints. The total reward at time step $t$ is composed of four components: safety ($R_{safe}$), navigation ($R_{nav}$), stability ($R_{stab}$), and terminal status ($R_{term}$).

$$R_t = \text{clip}(R_{safe} + R_{nav} + R_{stab} + R_{term}, R_{min}, R_{max})$$

where the total reward is clipped to the range $[R_{min}, R_{max}]$ (set to $[-500, 200]$) to stabilize training.

**Safety Reward ($R_{safe}$)** Safety is the primary objective. Penalties are applied for collisions and for triggering the safety shield intervention.

$$R_{safe} = \lambda_{col} \cdot \mathbb{I}_{collision} + \lambda_{shield} \cdot \mathbb{I}_{shield}$$

where $\mathbb{I}_{collision}$ is an indicator function that equals 1 if the aircraft violates the minimum separation requirements, otherwise 0. $\mathbb{I}_{shield}$ is an indicator function that equals 1 if the kinematic shield overrides the agent's action, otherwise 0.

**Navigation & Trajectory Reward ($R_{nav}$)** This component encourages the agent to follow the designated route and maintain efficient progress.

$$R_{nav} = R_{xtk} + R_{hdg} + R_{prog}$$

1. Cross-Track Error ($R_{xtk}$): Penalizes the deviation from the assigned flight path using both linear and quadratic terms to discourage large deviations.

$$R_{xtk} = \lambda_{xtk} \cdot d_{xtk} + \lambda_{xtk^2} \cdot (d_{xtk})^2$$

where $d_{xtk}$ is the perpendicular distance to the route segment.

2. Heading Alignment ($R_{hdg}$): Encourages the aircraft to align with the desired heading derived from a vector field guidance logic (which guides the aircraft back to the path).

$$R_{hdg} = \lambda_{hdg} \cdot \frac{|\Delta\psi|}{180°}$$

where $\Delta\psi$ is the difference between the current heading and the vector field desired heading.

3. Progress Reward ($R_{prog}$): Rewards movement towards the destination along the path and penalizes regression.

$$R_{prog} = \begin{cases} \min(\lambda_{prog} \cdot \Delta d, C_{prog\_max}) & \text{if } \Delta d > 0 \\ -\lambda_{regress} \cdot |\Delta d| & \text{if } \Delta d \leq 0 \end{cases}$$

where $\Delta d$ is the improvement in distance along the track compared to the previous step.

**Stability Reward ($R_{stab}$)** To ensure passenger comfort and flight feasibility, we penalize altitude deviation and abrupt control actions.

$$R_{stab} = \lambda_{alt} \cdot \frac{|h_t - h_{des}|}{100} + \lambda_{smooth} \cdot \frac{1}{N_a} \sum_{i=1}^{N_a} |a_t^{(i)}|$$

where $h_t$ is the current altitude and $h_{des}$ is the assigned (or target) altitude, both in feet. $a_t$ is the normalized continuous action vector output by the agent.

Terminal Reward ($R_{term}$) Sparse rewards are assigned upon episode termination based on the outcome.

$$R_{term} = \begin{cases} \lambda_{goal} & \text{if } d_{goal} < D_{arrival} \\ \lambda_{bound} & \text{if } d_{xtk} > D_{max\_width} \\ 0 & \text{otherwise} \end{cases}$$

where $D_{arrival}$ is the distance threshold to consider the goal reached, and $D_{max_width}$ is the maximum allowable cross-track error before episode termination.

# D. Architectural Details of SARL

Due to space constraints in the main paper, we provide a detailed elaboration on the structural details of the SARL framework in this appendix, offering a more comprehensive understanding of the network design.

## D.1. Actor and Critic

Within the SARL framework, the Actor Network and Critic Network adopt an independent architectural design with no shared parameters, as illustrated in Figure 9. Specifically, the Actor possesses its exclusive PERG module, which is utilized to encode local, egocentric observational information into latent representations $z_i$ for subsequent use by the SC-MoE decision head. Meanwhile, the Critic utilizes a separate, independent PERG encoder to process global states, thereby evaluating the value of joint actions. The rationale behind decoupling the representation encoders is to avoid gradient conflicts and non-stationarity issues during multi-agent training, as the learning pace of the Critic's value evaluation typically differs significantly from the Actor's policy updates. Further details of the PERG architecture are illustrated in Figure 10.

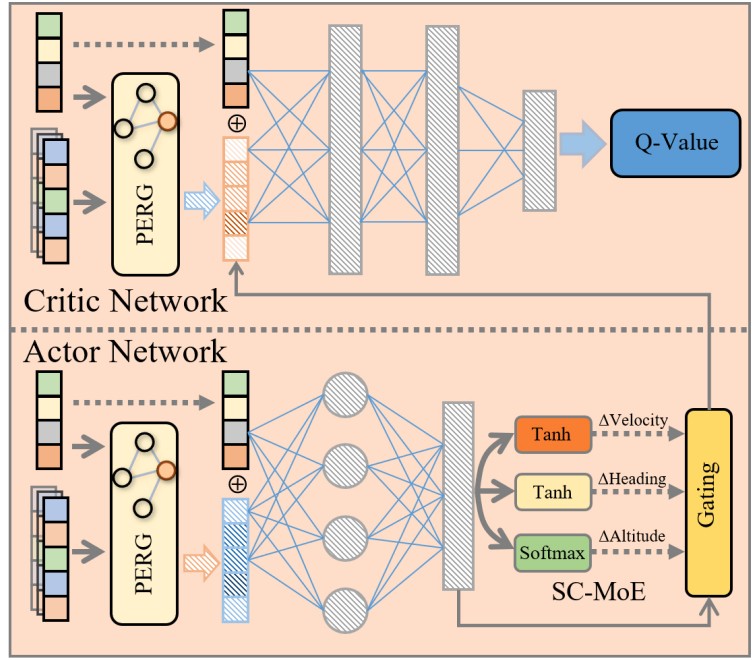

*Figure 9.* Architecture of Actor Network and Critic Network

**Critic Network Architecture** The primary function of the Critic network is to evaluate the Q-Value of multi-aircraft joint states and actions in high-density airspace.

1. Feature Input and Graph Representation: The Critic receives global state features. These features are first fed into the exclusive PERG module to extract global spatial topological relationships and high-order graph representations.

2. Feature Fusion: To preserve the original physical kinematic attributes, the network adopts a residual connection concept, concatenating the graph representation vectors extracted by PERG with the original state vectors along the channel dimension.

3. Value Evaluation: The concatenated fused features are input into a MLP backbone. As shown in the figure, this MLP consists of some fully connected hidden layers that progressively reduce the dimensionality of the high-dimensional features, ultimately outputting a scalar Q-Value to guide the Actor's policy updates.

**Actor Network Architecture** The Actor network is responsible for generating specific conflict resolution maneuvers based on the current aircraft's local perception. Its core lies in handling complex hybrid action spaces while enforcing operational sparsity.

1. Local Perception and Encoding: The Actor receives the agent's local observational state (including its own state and those of neighboring aircraft). Through an independent PERG module, the complex airspace interaction topology is encoded into environmental perception features.

2. Feature Fusion and Dimensionality Reduction: Similarly, the output features of PERG are concatenated with the original local state, and subsequently input into a single-hidden-layer MLP for initial feature extraction and mapping.

3. SC-MoE Action Output Head: Addressing the heterogeneity of 3D aircraft flight control, we design the Actor's output end as a SC-MoE architecture. The MLP's output features are simultaneously fed into four semantically specialized expert branches: Speed, Heading, Altitude, and a crucial "Hold" (Non-Op) expert to represent trajectory maintenance.

4. Sparse Gating Mechanism: To coordinate these maneuvers without inducing cross-dimensional control interference, the SC-MoE introduces a differentiable routing module utilizing Gumbel-Softmax. Instead of blending outputs, this router calculates gating logits and executes a hard selection. It adaptively determines which single maneuver dimension should dominate under the current situation, ultimately outputting a decoupled, sparse flight action command that better aligns with real-world Air Traffic Control protocols.

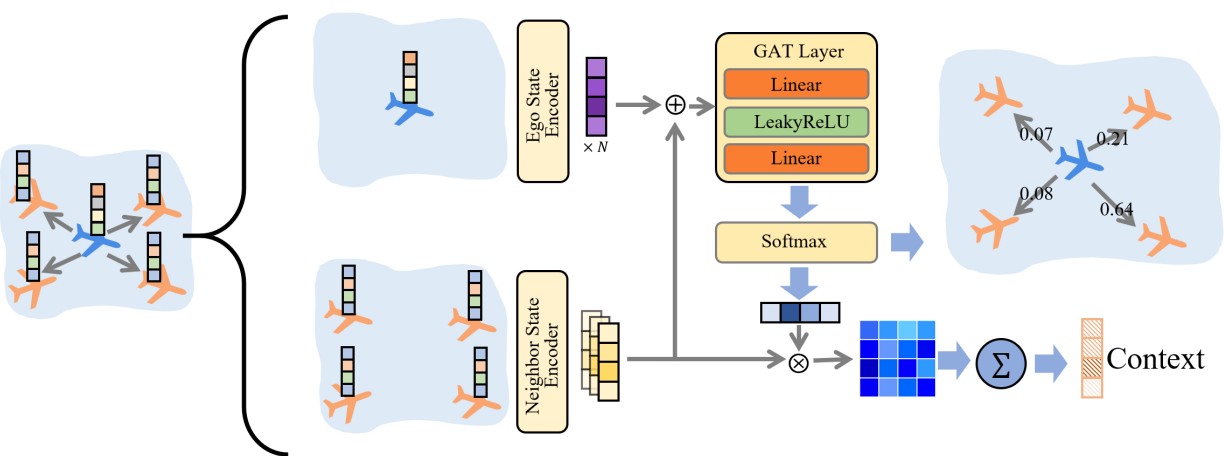

*Figure 10.* Architectural Details of PERG

## D.2. KSS Module

In this section, we introduce the working mechanism of KSS and its associated supplementary experiments. We elaborate on the operational details of KSS across three key aspects: trigger conditions, safety correction and priority, and multi-rule conflicts.

1. Trigger Conditions: The mechanism uses a constant velocity kinematic model to project the current state forward. A violation alert is triggered if the predicted distance at the Time to Closest Point of Approach is strictly less than the safety threshold and this event occurs within a predefined lookahead window.

2. Safety Correction and Priority: Upon triggering, the KSS overrides the RL agent's action with predefined evasive maneuvers. Following standard aviation heuristic rules, altitude adjustment is given the highest priority due to its rapid conflict resolution capability, followed by heading vector adjustments.

3. Multi-Rule Conflicts: In multi-agent scenarios with conflicting rules, the system adopts a decentralized "right-of-way" protocol (e.g., the aircraft on the right maintains its trajectory, while the yielding aircraft executes the KSS evasive maneuver).

Additionally, to further evaluate the performance impact of the KSS module, we conducted independent supplementary experiments. Table 4 demonstrates that this performance leap fundamentally stems from our neural network's structural optimizations—specifically, resolving dimensional mismatch and action coupling—rather than relying solely on the KSS as a safety net.

*Table 4.* Supplementary experiments on the performance impact of the KSS module.

| Method | Succ Rate (%) | Avg Op. Cost | Avg Min Sep |
| --- | --- | --- | --- |
| MADDPG | 94.84 | 345.56 | 17.32 |
| MADDPG w/ KSS | 97.11 | 525.52 | 19.65 |
| MAPPO | 86.12 | 512.15 | 11.98 |
| MAPPO w/ KSS | 93.50 | 945.07 | 17.27 |
| SARL | **99.28** | **157.54** | **24.39** |

# E. Supplementary Experiments

In this section, we provide extensive supplementary experiments to verify the scalability and robustness of SARL.

## E.1. Scalability via Scenario Transfer

To evaluate the generalization ability of the model across different scenarios, we train the model in `Train_Case` and test it in three `Eval_Case` environments with varying traffic conditions. The experimental results, as shown in Table 5, demonstrate that our model achieves superior performance across diverse scenarios, exhibiting excellent scenario generalization capabilities.

*Table 5.* Scalability via scenario transfer.

| Environment | Succ Rate (%) | Avg Op. Cost | Avg Min Sep |
| --- | --- | --- | --- |
| Train_Case | 99.28 | 157.54 | 24.39 |
| Eval_Case1 | 99.14 | 201.80 | 22.82 |
| Eval_Case2 | 99.33 | 198.12 | 25.29 |
| Eval_Case3 | 99.23 | 153.22 | 27.00 |

## E.2. Scalability in High-Density Scenarios

To verify the scalability of the model in high-density scenarios, we increased the number of aircraft in the airspace to 30 (the simultaneous presence of up to 30 aircraft within an airspace sector represents an extremely high-density and highly complex environment). Table 6 demonstrates that our model exhibits superior performance across various density scenarios and maintains this robust performance even in extremely high-density environments.

*Table 6.* Scalability in high-density scenarios.

| Max Nums | Succ Rate (%) | Avg Op. Cost | Avg Min Sep |
| --- | --- | --- | --- |
| 6 | 99.79 | 156.32 | 38.68 |
| 12 | 98.58 | 154.23 | 30.79 |
| 18 | 98.38 | 152.76 | 26.67 |
| 24 | 99.17 | 153.70 | 24.92 |
| 30 | 99.29 | 157.94 | 22.78 |

## E.3. Robustness to Noise and Latency

This experiment simulates scenarios of degraded information transmission quality by injecting varying degrees of noise into the communication links. The performance metrics for conflict resolution under different Signal-to-Noise Ratio (SNR) conditions are presented in Table 7. The experimental results indicate that SARL possesses strong robustness to channel noise. As the SNR decreases, the conflict resolution success rate of SARL maintains strong performance, and the average minimum separation does not exhibit a precipitous decline in safety guarantees. Compared to the noise-free scenario, the fluctuation range of the success rate under different SNR conditions is extremely narrow, indicating that the model does not overly rely on idealized, precise position data.

*Table 7.* Robustness to communication noise.

| SNR | Succ Rate (%) | Avg Op. Cost | Avg Min Sep |
|-----|---------------|--------------|-------------|
| 10 | 99.03 | 157.33 | 24.46 |
| 20 | 99.29 | 151.75 | 22.82 |
| 30 | 99.30 | 158.29 | 24.17 |
| 40 | 99.23 | 153.54 | 24.82 |
| None | 99.28 | 157.54 | 24.39 |

### E.4. Robustness to Information Latency

To verify the fault tolerance of the algorithm in the temporal dimension, this experiment simulates situations where there is a lag in the transmission of position and intent information. The performance under different latency conditions is detailed in Table 8. The data reveals that as information latency increases, the average minimum separation exhibits a generally increasing trend. This indicates that in situations with delayed perceptual information, the model tends to adopt a more conservative resolution strategy, compensating for the uncertainty in state prediction by actively increasing the distance between aircraft. Although latency causes slight fluctuations in the success rate, the framework still maintains excellent overall performance.

*Table 8.* Robustness to information latency.

| Latency (s) | Succ Rate (%) | Avg Op. Cost | Avg Min Sep |
|-------------|---------------|--------------|-------------|
| 0.0 | 99.28 | 157.54 | 24.39 |
| 0.05 | 98.90 | 152.12 | 26.52 |
| 0.10 | 98.80 | 152.89 | 26.83 |
| 0.20 | 98.71 | 153.63 | 27.65 |
| 0.50 | 99.10 | 151.70 | 27.57 |
| 1.00 | 98.67 | 151.45 | 27.91 |

