# OpenReview forum: "SARL: Structure-Aligned Reinforcement Learning for Bridging the Perception-Action Gap in Airspace"
_ICML.cc/2026/Conference — ICML 2026 regular_

### Official Review · Reviewer_nwYL · 2026-03-10

**Soundness:** 2
**Presentation:** 3
**Significance:** 2
**Originality:** 2
**Overall Recommendation:** 3
**Confidence:** 4

**Summary:**

This paper proposes a novel reinforcement learning framework, SARL, to address perception scalability and action stability challenges in the automated aircraft scheduling problem. Specifically, SARL incorporates a graph neural network–based encoder to process information from a variable number of aircraft, a mixture-of-experts model to enhance policy stability, and a kinematic safety layer to improve operational safety. Experimental results in a simulated environment demonstrate improved performance in terms of success rate and average cost compared to several baseline methods.

**Compliance With Llm Reviewing Policy:**

Affirmed.

**Final Justification:**

After the discussion with the authors during the rebuttal phase, my understanding of the work has improved and some concerns have been clarified. However, a few limitations remain inadequately addressed, particularly regarding the approach's generality. My current rating reflects my final assessment.

**Key Questions For Authors:**

1. Definition of the Perception–Action Gap. The paper seems to address two issues separately: scalability in perception as the number of aircraft increases, and stability in action selection. I am somewhat confused by the term perception–action gap, which suggests a connection or mismatch between perception and action. However, this relationship is not clearly introduced or formalized in the paper. The notion of misalignment between perception and action is also unclear. Could the authors clarify what specific type of misalignment they refer to?

2. Reward formulation in the multi-agent setting. In the multi-agent setup, the agents are trained to maximize a reward defined for the entire environment rather than an individual reward. If I understand correctly, the authors state that the reward model is shared among all agents, but it is not clearly explained how the global reward is defined or aggregated. More details on the reward design would help clarify the training setup.

3. Problem scale and comparison with traditional methods. From Figure 4, the problem instance appears relatively small and may potentially be solvable using traditional conflict-resolution methods. Is there a reason why classical approaches are not included as baselines for comparison?

4. Graph architecture description. The use of a graph representation and graph attention mechanisms to process relational information is interesting. It would be helpful to include a figure illustrating the architecture of the physics-encoded relational graph and the main operations described in Section 3.1, as this would make the design easier to understand.

5. Contribution of individual components. The proposed method consists of three components. However, it is not entirely clear which component contributes most significantly to the overall performance. From Table 2, it appears that the safety layer greatly improves the performance of the architecture and may be the dominant component. It would therefore be helpful to add the KSS layer to the baseline methods to evaluate whether the observed improvement still persists.

6. Observation size vs. performance (Figure 5). In Figure 5, the success rate for SARL is constant as the observation size changes. This result suggests that the agent may not need to observe neighboring agents. Intuitively, one might expect that observing more neighbors (larger 𝑘) would improve performance. Could the authors provide an explanation for this behavior?

7. Presentation issues. In the abstract: Kinematic Kafety Shield → likely Kinematic Safety Shield. ; Equation (11): Dncoder → likely Decoder.

**Limitations:**

See the weaknesses.

**Strengths And Weaknesses:**

Strengths

1. The research problem is interesting, and the proposed solution appears technically sound.
2. The empirical results provide sufficient evidence of the improvement of the proposed method over several baselines.
3. The paper is generally well presented and easy to follow.

Weaknesses

1. The main innovations are highly application-dependent, and the proposed solution is largely limited to this specific problem. As a result, the contribution to the broader machine learning community is limited.
2. Some technical terms are not clearly defined, and some technical descriptions require further clarification.
3. The justification for some technical components is missing or insufficiently discussed.
4. There are some minor presentation issues.

Please see the detailed questions below for further comments.

---

> ### Author Rebuttal · Authors · 2026-03-30
>
> We sincerely thank the reviewer for their valuable feedback and for pointing out the typographical errors in our manuscript. We have carefully addressed your comments and provide our detailed responses below:
>
> 1. **Clarification on "Perception-Action Gap" and "Mismatch"**
>    We sincerely thank the reviewer for highlighting the potential ambiguity regarding this concept. The term "Mismatch" refers to the structural incongruity between standard RL neural network architectures and the physical realities of the ATC domain. This misalignment specifically manifests as a "dual gap" (at both the perception end and the action end individually), rather than a gap between perception and action within aviation RL itself. For specific details, please refer to the paragraph starting at line 58.
>
> 2. **Multi-Agent Reward Design**
>    We appreciate the opportunity to clarify the formulation of our multi-agent reward mechanism. In our framework, we adopt the Centralized Training with Decentralized Execution paradigm. During the training phase, this Critic evaluates the joint state and joint actions of all agents to estimate a centralized Q-value. This mechanism ensures that each agent's policy gradient update fully accounts for global environmental information, thereby fostering emergent, implicit collaborative behaviors. For detailed formulations of the reward functions, please refer to Appendix C.2.
>
> 3. **Problem Scale and Traditional Baselines**
>    We wish to contextualize our problem scale. While a scenario involving over 30 aircraft might appear visually sparse compared to UAV swarm simulations, in the domain of civil aviation, accommodating more than 30 aircraft simultaneously within a single airspace sector represents an environment of extremely high density and complexity.Furthermore, although traditional geometric or potential field-based solvers (such as ORCA or MVP) serve as classic and efficient baselines for general multi-agent collision avoidance, they were excluded from our primary evaluation scope due to their fundamental mismatch with the strict, domain-specific operational constraints and kinematic limitations of ATC.
>
> 4. **Detailed Illustration of PERG**
>    We greatly appreciate your positive feedback regarding this concept. To make PERG's architecture and information flow more intuitive, we have added a dedicated sub-figure (https://anonymous.4open.science/r/Rebuttal-6BC9/PERG.png) that details its internal mechanisms.
>
> 5. **Component Contributions and Integrating KSS into Baselines**
>    Our PERG and SC-MoE modules are primarily designed to structurally align the RL framework with the aviation domain, whereas the KSS functions to guide the model in learning effective, compliant policies. Importantly, the overall performance enhancement is not solely attributable to the KSS. As demonstrated in the ablation study (Figure 5a), the model's performance already surpasses the baseline simply with the introduction of the PERG module. Furthermore, acting on your constructive suggestion, we conducted an additional experiment evaluating MAPPO/MADDPG + KSS. The results indicate that while forcibly integrating the KSS into MAPPO/MADDPG yields some performance improvements, the baseline algorithm ultimately fails to autonomously learn the safety boundaries. In stark contrast, SARL demonstrates exceptional adaptability to the aviation domain through the organic integration of its specialized modules within the RL paradigm.
> | Method | Succ Rate (%) | Avg Op. Cost | Avg Min Sep |
> |--------|---------------|--------------|-------------|
> | MADDPG | 94.84 | 345.56 | 17.32 |
> | MADDPG w KSS | 97.11 | 525.52 | 19.65 |
> | MAPPO | 86.12 | 512.15 | 11.98 |
> | MAPPO w KSS | 93.50 | 945.07 | 17.27 |
> | **SARL** | **99.28** | **157.54** | **24.39** |
>
> 6. **Relationship Between Observation Scale and Performance**
>    We sincerely apologize for any confusion caused by the presentation of Figure 5. The fact that SARL's performance curve is depicted as a constant horizontal line does not imply that the algorithm ignores neighboring nodes. In Figure 5, the horizontal axis represents the artificially imposed, fixed number of neighbors ($k$) specifically required by the baseline model (which utilizes an MLP). For such MLP models, hyperparameter tuning of $k$ is a mandatory step. Conversely, SARL utilizes the PERG module, which dynamically processes all valid neighbors within its physical detection radius, thereby entirely circumventing the need to predefine a fixed $k$ value. Consequently, we plotted SARL's performance as a horizontal reference line to clearly demonstrate a key finding: regardless of how extensively the fixed $k$ value of the baseline model is fine-tuned, the dynamic graph perception capability inherent to SARL consistently achieves superior performance.

---

> > ### Author Rebuttal · Reviewer_nwYL · 2026-04-02
> >
> > I appreciate the authors’ effort in addressing my concerns. The clarity of several key aspects, such as the perception–action gap, reward design, and result interpretation, has improved significantly.
> >
> > However, I still find that the paper primarily integrates several existing approaches and applies them to a specific automated aircraft scheduling problem, leveraging domain knowledge to achieve strong performance. While this is valuable from an application perspective, it remains unclear how the proposed method advances machine learning more broadly or generalizes to other domains.
> >
> > In its current form, the work appears more aligned with a domain-specific robotics or operations-focused contribution rather than offering general methodological insights. Given the limited broader impact, I would maintain my current rating.

---

> > > ### Author Response · Authors · 2026-04-07
> > >
> > > Thank you for your continued engagement and for acknowledging the improvements in our revision.
> > >
> > > While we deeply respect your perspective and final evaluation, we feel a sense of regret that we could not fully convince you of the broader methodological implications of our work. We genuinely believe that the architectural principles of structural alignment (e.g., mitigating dimensionality mismatch and action coupling) have significant potential to inspire other constrained embodied AI domains beyond aviation.
> > >
> > > Nevertheless, your rigorous and constructive feedback has undeniably made our paper stronger. We sincerely appreciate the time and effort you have dedicated to this review process.

---

### Official Review · Reviewer_RT6E · 2026-03-13

**Soundness:** 3
**Presentation:** 2
**Significance:** 2
**Originality:** 2
**Overall Recommendation:** 3
**Confidence:** 4

**Summary:**

This paper proposes a new approach, called SARL, to address the structural misalignment between model architectures and domain requirements in Air Traffic Management. Experiments show some outperforming results over several baselines.

**Compliance With Llm Reviewing Policy:**

Affirmed.

**Final Justification:**

I am very grateful to the authors for the clarification, but a major problem with the manuscript is that the comparative experiments and scalability analysis are still insufficient. Therefore, I will stick to my decision.

**Key Questions For Authors:**

Please see the pros and cons.

**Limitations:**

Please see the pros and cons regarding the generalization.

**Strengths And Weaknesses:**

Soundness,

This paper is well-motivated and technically sound.


Representation:

The description is somewhat like "AI-generated". For example, what do you mean by "unique characteristics" (Line 18)? "model architectures and domain requirements" should be specifically elaborated(Lines 19-23). How to understand "decoupling" (Line 79)?

I think the authors should carefully check this and clarify by their own understanding.

Significance:

1. In the experiment section, the meaning of the action "Compliance" and the action "Safety" requires further explanation and clearer differentiation. Specifically, the authors need to justify why "sparsity and smoothness" can be strictly corresponded to "Compliance".
2. To further demonstrate perception scalability, the experimental settings need to be refined, particularly by introducing scenarios that observe the dynamic changes in the neighborhood information.
3. While the appendix provides a Dec-POMDP formulation, Figure 2 illustrates the framework; it is still unclear how and where the "partial observability" is actually reflected in the design. The authors need to resolve this discrepancy and ensure consistency between the theoretical formulation and the engineering implementation.

Originality:

Although the logic is clear, the technical novelty is insufficient, which is reflected in the following aspects:

1. GNNs have been widely used in MARL. The paper needs to sufficiently highlight the significant distinctions or unique advantages of the proposed method compared to existing GNN-based MARL frameworks.

2. The SC-MoE and KSS components are domain-specific, resulting in a lack of generalizability (extensibility) to other domains.

---

> ### Author Rebuttal · Authors · 2026-03-30
>
> We sincerely thank the reviewer for their valuable feedback and for providing us the opportunity to clarify these aspects of our work. Please find our detailed responses below:
>
> 1. **Clarification of Terminology**
>    We apologize for any misunderstandings caused by the brevity of certain explanations in the original manuscript.
>    - **"unique characteristics"** is a broad encapsulation of the strict operational constraints inherent in real-world civil aviation. According to standard ATC manuals, controllers strictly avoid issuing simultaneous, multi-dimensional commands (e.g., "descend 1000 feet, turn left 30 degrees, and reduce speed by 50 knots"), nor do they issue continuous, high-frequency adjustments that cause severe trajectory oscillations. This corresponds directly to the sparsity and smoothness emphasized in our paper.
>    - **"model architectures and domain requirements"** refers to the core problem our work addresses: standard RL models output dense, coupled continuous actions that inherently conflict with the aforementioned aviation domain requirements (as detailed from Line 58 onwards).
>    - **"decoupling"** refers to the decomposition of standard multi-dimensional RL outputs. Outputting a coupled continuous vector (e.g., $[a_{speed}, a_{heading}, a_{altitude}]$) simultaneously demands pilot adjustments across all axes, leading to cognitive overload and potential operational errors. To resolve this, we decompose the continuous multi-dimensional action space into strictly orthogonal operational primitives. At any given time step, the agent selects only one semantic expert (e.g., exclusively adjusting heading) while maintaining the other dimensions constant, thereby eliminating cross-dimensional interference.
>
> 2. **Compliance vs. Safety**
>    In our framework, Safety serves as the absolute baseline; it strictly measures whether a collision or Loss of Separation has occurred. However, achieving theoretical safety does not guarantee that the generated maneuvers are practically executable. To be viable for real-world deployment, the policy must also demonstrate Compliance—meaning it must adhere to the "unique characteristics" of standard ATC directives mentioned above. Therefore, we utilize sparsity (intervening only when absolutely necessary, executed via the "Hold" expert) and smoothness (executing single-dimensional maneuvers without oscillation) as precise, mathematical proxy metrics to quantify ATC compliance.
>
> 3. **Scalability and Dynamic Neighborhoods**
>    We wish to clarify that our current experimental setup inherently features highly dynamic neighborhood topologies. In our BlueSky simulation, the interaction graph is strictly non-static. Because the PERG interaction graph is constructed based on radar separation standards—a fundamentally distance-based control paradigm—the number of observed neighbors fluctuates dynamically. At any given time step, an agent can exclusively observe other aircraft that enter its specific spatial radius. This dynamically varying neighborhood perfectly validates our claims regarding the PERG module's structural scalability and its ability to handle partial observability.
>
> 4. **Module Novelty and Generalization Capability**
>    Unlike standard Graph Neural Networks in the MARL domain, which primarily focus on learning implicit communication protocols, our PERG module explicitly embeds physical inductive biases directly into both the graph construction and the node feature initialization processes. Compared to generic GNNs that must learn spatial physics entirely from scratch, this physics-grounded approach drastically accelerates training convergence. Although SARL is contextualized within the aviation domain, its core architecture is fundamentally a universal framework suitable for any high-density spatial navigation task involving continuous states and decoupled action constraints. Consequently, this framework is highly generalizable and can be readily extended to domains including, but not limited to, autonomous vehicle platooning, multi-robot warehouse logistics, and UAV swarms.

---

> > ### Author Rebuttal · Reviewer_RT6E · 2026-04-02
> >
> > I am very grateful to the authors for the clarification, but a major problem with the paper is that the comparative experiments and scalability analysis are still insufficient. Therefore, I will stick to my decision.

---

> > > ### Author Response · Authors · 2026-04-07
> > >
> > > Thank you for your prompt reply and for sharing your final evaluation. We deeply respect your rigorous standards regarding the comparative and scalability analysis. Because we take your critique very seriously, we have fully utilized the remaining discussion window to conduct comprehensive, extended stress-tests. We are excited to share these newly generated results with you below, which we believe definitively prove the scalability and superiority of the SARL framework.
> > >
> > > 1. **Scalability Analysis**
> > >    To explicitly address the scalability concern, we extended the environment to an high-density setting. we increased the number of aircraft in the airspace to 30 (the simultaneous presence of over 30 aircraft within an airspace sector represents an airspace environment of extremely high density and complexity). The table below demonstrates that our model exhibits superior performance across various density scenarios—a level of performance that it successfully maintains even within high-density environments.
> > >
> > > | max_num | Succ Rate (%) | Avg Op. Cost | Avg Min Sep |
> > > |------------|---------------|--------------|-------------|
> > > | 6          | 99.79         | 156.32       | 38.68       |
> > > | 12         | 98.58         | 154.23       | 30.79       |
> > > | 18         | 98.38         | 152.76       | 26.67       |
> > > | 24         | 99.17         | 153.70       | 24.92       |
> > > | 30         | 99.29         | 157.94       | 22.78       |
> > >
> > > 2. **Scalability via Scenario Transfer (Zero-Shot Generalization)**
> > >    Furthermore, we tested the model's scalability across entirely unseen airspace topologies. The experimental results demonstrate that our model maintains superior performance within the zero-shot space, thereby validating the universal scalability of its learned sparse strategies and physical constraints—rather than merely overfitting to specific simulation instances.
> > >
> > > | Env          | Succ Rate (%) | Avg Op. Cost | Avg Min Sep |
> > > |--------------|---------------|--------------|-------------|
> > > | Train_Case   | 99.28         | 157.54       | 24.39       |
> > > | Eval_Case1   | 99.14         | 201.80       | 22.82       |
> > > | Eval_Case2   | 99.33         | 198.12       | 25.29       |
> > > | Eval_Case3   | 99.23         | 153.22       | 27.00       |
> > >
> > > These empirical results explicitly address the boundaries of our comparative and scalability evaluations. We hope this concrete evidence demonstrates our commitment to your rigorous standards and fully alleviates your remaining concerns.

---

### Official Review · Reviewer_QXX5 · 2026-03-13

**Soundness:** 2
**Presentation:** 3
**Significance:** 2
**Originality:** 2
**Overall Recommendation:** 4
**Confidence:** 2

**Summary:**

The paper shows structure-aligned reinforcement learning for aircraft control tasks in multi-agent airspace environments. The framework aims to incorporate domain structure into the RL architecture to better model interactions between aircraft and environment dynamics. It proposes a PERG module to capture structured airspace relationships among aircraft, then the state encoder and graph attention network process spatial interactions between agents. The structure-conditioned mixture-of-experts is applied to generate control actions aligned with different control dimensions: velocity, heading, and altitude. Besides, the Kinematic Safety Shield (KSS) is also used to enforce safe actions by projecting raw actions into a safe action space. In general, the authors aim to improve learning efficiency and safety by embedding domain structure into both the representation and action generation process.

**Compliance With Llm Reviewing Policy:**

Affirmed.

**Final Justification:**

Thanks for the rebuttal; part of my concerns were addressed. I would adjust the score based on the merit of the work.

**Key Questions For Authors:**

1. The PERG module is used to construct the interaction graph among aircraft agents before applying the GAT layer.  Could the authors clarify on how neighboring aircraft are selected (e.g., distance threshold, k-nearest neighbors, or other rules)? and i wonder whether the interaction graph changes dynamically during simulation?

2. In Figure 2, the actor and critic networks appear within the same visual block and share similar colors, which makes it somewhat difficult to distinguish their structural separation. Could the authors clarify how the representations are shared between the actor and critic networks, and whether separate encoders were considered?

3. The method requires the aircraft state, such as position, speed, etc, to construct the graph and compute relational attention. In real air traffic systems, such observations may contain noise or delays. Could the authors explain if it's possible to consider how robust SARL is to noisy or delayed state observations, and whether performance degrades significantly under such conditions? And what future work can be done in this direction?

**Limitations:**

There is no clear limitation section in the paper,  so some limitations can be considered to discuss, such as the Sim-to-real deployment gap: the evaluation is conducted entirely in the BlueSky simulation, however, real-world airspace operations involve additional complexities: sensor noise and communication delays. It's also worthwhile to understand the scalability in high-density airspace.

**Strengths And Weaknesses:**

### Strength:

1. The paper clearly motivates the problem setting of aircraft control and explains why incorporating structural priors into reinforcement learning architectures can be beneficial. The airspace control and trajectory planning problems are important application domains for reinforcement learning, especially in safety-critical systems. The idea of incorporating domain structural priors into RL architectures is meaningful.

2. The proposed framework is described in a structured and modular manner and the paper introduces each component (PERG, GAT, SC-MoE, and KSS) sequentially, which helps readers follow the design rationale behind the architecture.

3. Safety-aware action mechanism is important: The Kinematic Safety Shield (KSS) provides a practical mechanism to enforce safety constraints on actions. Such a post-processing safety filter is a reasonable design for safety-critical control systems and aligns with safe RL literature.

### Weakness:
1. In the SC-MoE module, it introduces a structure-conditioned mixture-of-experts architecture to generate control outputs for velocity, heading, and altitude. However, the paper does not analyze how the experts are actually utilized during training. Since MoE models are known to suffer from issues such as: expert collapse and uneven expert utilization,  thus without reporting statistics such as expert usage distribution, it is difficult to verify whether the MoE mechanism truly improves representation capacity or simply behaves like a standard MLP.

2. While this design of graph attention layers is suitable for structured airspace environments, the paper does not seem to analyze how the approach scales as the number of aircraft increases. In dense airspace scenarios, the number of interaction edges may grow rapidly, which could increase both computational cost and attention instability. A discussion or experiment evaluating performance under varying traffic densities would help assess the scalability of the approach.

3. From the Figure 2, the PERG, state encoder, and GAT layers appear to produce shared representations that are used by both the actor and critic networks. However, the paper does not discuss whether this shared representation introduces optimization interference between policy and value learning. In actor–critic methods, representation sharing can sometimes destabilize training when gradients from value estimation conflict with policy updates. It would be interesting to see an ablation comparing shared vs separate representations.

---

> ### Author Rebuttal · Authors · 2026-03-30
>
> We sincerely thank the reviewer for the thoughtful review and valuable feedback. We address your specific concerns below:
>
> 1. **Considerations regarding SC-MoE and Expert Collapse**
>    We acknowledge that expert collapse and routing imbalance are classic challenges in standard MoE architectures. However, in our SC-MoE model, the experts are heterogeneous and grounded in distinct physical semantics. In real-world ATC scenarios, the distribution of tactical interventions is inherently unbalanced:
>    - **"Hold"** (maintaining the current state) is the default;
>    - **"Heading"** vectoring is the most frequently used method for conflict resolution;
>    - **"Altitude"** and **"Speed"** adjustments are used sparingly due to considerations for passenger comfort and aircraft kinematic inertia.
>
>    Therefore, the "imbalance" in expert utilization observed in our model is not a pathological expert collapse, but rather strong empirical evidence that the routing module has successfully learned the domain-specific physical constraints and operational protocols of ATC. Furthermore, to ensure sufficient training for all experts and prevent premature routing collapse during the initial exploration phase, we employ a Gumbel-Softmax sampling mechanism coupled with temperature annealing. In the early stages of training, a higher temperature parameter forces the routing module to smoothly and thoroughly explore all physically meaningful experts. In the later stages, a lower temperature ensures the model strictly adheres to the action sparsity required in ATC scenarios. The specific action distributions can be referenced in Figure 5(b).
>
> 2. **PERG Graph Construction and Dynamic Topology**
>    First, the construction of our PERG interaction graph relies on radar-based separation criteria, which are widely used in aviation control and fundamentally rely on a distance-based threshold. Consequently, our interaction graph is highly dynamic during simulation. At each simulation time step, edges are added or removed in real-time as the aircraft move in and out of each other's detection radii. The ability to seamlessly process this dynamically changing topological structure is precisely the core advantage of the PERG module.
>
> 3. **Actor-Critic Architecture and Representation Sharing**
>    We apologize for any visual ambiguity in Figure 2 and appreciate the opportunity to clarify the network architecture. In the SARL framework, the Actor and Critic networks utilize completely independent encoders and do not share parameters. Specifically, the Actor has its own dedicated PERG module to encode local, ego-centric observations into latent representations $z_i$, which are then utilized by the SC-MoE decision head. Meanwhile, the Critic employs a separate, independent encoder to process the global state and evaluate the joint action value. We deliberately chose not to share representation encoders to avoid gradient conflicts and non-stationarity issues during training, as the Critic and Actor typically learn at different paces. To eliminate this visual confusion, we will provide a more detailed and distinct architectural diagram in the revised manuscript(https://anonymous.4open.science/r/Rebuttal-6BC9/Actor-Critic.png).
>
> 4. **Robustness to Noise and Latency**
>    The SARL architecture inherently possesses a degree of tolerance to observation noise. The internal attention mechanism within the PERG module dynamically assigns weights to neighboring targets. If a neighbor's kinematic state exhibits sharp fluctuations due to noise interference, the attention mechanism helps smooth out and filter these anomalous features by assigning them lower weights. To address your concern directly, we conducted an additional experiment where we injected random noise into the state observations; the results, demonstrating the model's resilience, are presented in the table below.
> | SNR | Succ Rate (%) | Avg Op. Cost | Avg Min Sep |
> |-----|---------------|--------------|-------------|
> | 10  | 99.03         | 157.33       | 24.46       |
> | 20  | 99.29         | 151.75       | 22.82       |
> | 30  | 99.30         | 158.29       | 24.17       |
> | 40  | 99.23         | 153.54       | 24.82       |
> | None| 99.28         | 157.54       | 24.39       |
>
>    We acknowledge, however, that severe state delays or extreme noise interference will inevitably degrade the optimality of our sparse routing decisions. In future work, we plan to address this by incorporating Domain Randomization (e.g., injecting Gaussian noise during training) and formulating the problem as a Partially Observable Markov Decision Process (POMDP), utilizing historical sequences or RNN/Transformer architectures to explicitly handle latency.

---

> > ### Author Rebuttal · Reviewer_QXX5 · 2026-04-01
> >
> > Thank you for the rebuttal and clarifications.
> >
> > The authors addressed several concerns, for example: the clarification of the actor–critic architecture, the dynamic graph construction in PERG, and the robustness experiment under observation noise.
> >
> > However, some of my concerns remain only partially addressed.
> >
> > 1. Regarding the SC-MoE module, while the explanation of domain-driven imbalance is reasonable, the rebuttal does not provide concrete evidence (e.g., expert utilization statistics or routing behavior over training) to verify that the model avoids expert collapse and leverages multiple experts.
> >
> > 2. Although the construction of the PERG graph is clarified, the scalability concern under high-density airspace remains insufficiently explored.
> >
> > 3. The noise robustness is partially addressed, but the discussion of robustness under delayed or partially observed states remains at the level of future work.
> >
> > Overall, while the rebuttal improves clarity and addresses some points, several core concerns regarding mechanism validation and scalability remain open.

---

> > > ### Author Response · Authors · 2026-04-07
> > >
> > > We sincerely thank you for your constructive feedback. We fully agree that theoretical explanations must be backed by concrete empirical evidence. In this final response, we have conducted additional experiments to provide the exact statistics and stress-test results you requested.
> > >
> > > 1. **Concrete Evidence Against Expert Collapse in SC-MoE**
> > > To verify whether SC-MoE is indeed functioning as intended—and to confirm the absence of the "multi-expert collapse" problem—we explicitly logged and displayed the action frequencies of individual experts across the entire training batch. As shown in the table below, during the initial stages of training, changes in heading angle predominated; this is consistent with aviation regulations, which prioritize the use of heading adjustments for collision avoidance when separation distances meet the required criteria. During the intermediate stages, the distribution became more balanced as the aircraft explored and discovered a variety of alternative maneuvers. In the final stages, "no-action" instances became the majority, indicating that the flight path had stabilized. The performance of the router at each stage aligned with our theoretical expectations, thereby confirming that SC-MoE did not suffer from the multi-expert collapse problem.
> > >
> > > | Episode | Non_Op | Spd_Op | Alt_Op | Hdg_Op |
> > > |---------|--------|--------|--------|--------|
> > > | 10      | 33.4%  | 25.0%  | 16.6%  | 25.0%  |
> > > | 100     | 13.2%  | 32.7%  | 22.6%  | 31.5%  |
> > > | 500     | 8.1%   | 16.3%  | 20.8%  | 54.8%  |
> > > | 800     | 52.2%  | 13.2%  | 14.5%  | 20.1%  |
> > > | 1000    | 67.3%  | 12.4%  | 6.3%   | 14.0%  |
> > >
> > > 2. **Scalability in High-Density Scenarios**
> > > To validate the scalability of our model in high-density scenarios, we increased the number of aircraft in the airspace to 30 (the simultaneous presence of over 30 aircraft within an airspace sector represents an airspace environment of extremely high density and complexity). The table below demonstrates that our model exhibits superior performance across various density scenarios，a level of performance that it successfully maintains even within high-density environments.
> > >
> > > | max_num | Succ Rate (%) | Avg Op. Cost | Avg Min Sep |
> > > |---------|---------------|--------------|-------------|
> > > | 6       | 99.79         | 156.32       | 38.68       |
> > > | 12      | 98.58         | 154.23       | 30.79       |
> > > | 18      | 98.38         | 152.76       | 26.67       |
> > > | 24      | 99.17         | 153.70       | 24.92       |
> > > | 30      | 99.29         | 157.94       | 22.78       |
> > >
> > > 3. **Robustness to Information Latency**
> > > To address the issue of latency, we introduced varying time delays to validate our model's performance. The table below demonstrates that our model maintains superior performance even in the presence of information delays as long as one second, thereby proving its high robustness and foresight—specifically, its ability to effectively mitigate the impact of information latency to a significant degree.
> > >
> > > | Delay_time(s) | Succ Rate (%) | Avg Op. Cost | Avg Min Sep |
> > > |---------------|---------------|--------------|-------------|
> > > | 0.0           | 99.28         | 157.54       | 24.39       |
> > > | 0.05          | 98.90         | 152.12       | 26.52       |
> > > | 0.1           | 98.80         | 152.89       | 26.83       |
> > > | 0.2           | 98.71         | 153.63       | 27.65       |
> > > | 0.5           | 99.10         | 151.70       | 27.57       |
> > > | 1.0           | 98.67         | 151.45       | 27.91       |

---

### Official Review · Reviewer_oGNh · 2026-03-22

**Soundness:** 3
**Presentation:** 2
**Significance:** 3
**Originality:** 2
**Overall Recommendation:** 5
**Confidence:** 3

**Summary:**

SARL is a multi-agent reinforcement learning approach for aircraft conflict resolution that fixes a key mismatch between how models perceive airspace and generate actions. It uses a graph-based encoder for scalable perception, a sparse mixture-of-experts policy for realistic control decisions, and a rule-based safety shield to enforce aviation constraints. Experiments show it achieves higher safety and efficiency than existing methods while producing more stable and interpretable behavior.

**Compliance With Llm Reviewing Policy:**

Affirmed.

**Final Justification:**

I apologize for the delay. Thanks the authors for the comment, I will slightly improve my score.

**Key Questions For Authors:**

-It is not entirely clear how the multi-agent aspect is fully leveraged. While interactions between aircraft are modeled through the graph-based perception module, it remains ambiguous whether action planning is truly coordinated across agents. Can you explain it better?
- Do you plan an adversarial scenario, where not all the aircraft can be controlled?
- Is this approach broadly applicable to other scenario, not in the Air traffic subject?

**Limitations:**

- Limited Experiments, to one single kinematic model, and aircraft behaviour
- Potential Overfitting to Scenario Design
- While PERG improves input scalability, the architecture may introduce significant computational overhead

**Strengths And Weaknesses:**

## Strengths
### Practical Approach
Structuring the approach of Traffic Control, dividing it into perception, control and safety guards, it represents a safer and practical approach into such safety-critical environment.
### Architectural Soundness
The usage of a Graph network embedding the neighbourogh feature combined with ego ones, abstract the causality of possible collisions in a structured way.


## Weaknesses
### Limited Evaluation
The result are compared in an environment restricted to a single kinematic model, to non interactive other vehicles, and the multi-agent approach is not fully exploited.
### Partial Reliance on Rule
While the safety shield improves reliability, it introduces hand-crafted rules that reduce the level of end-to-end learning and may limit adaptability to unseen or evolving regulations.

---

> ### Author Rebuttal · Authors · 2026-03-30
>
> We sincerely appreciate your feedback and the recognition of our work. Please find our responses to your specific questions below:
>
> 1. **Multi-Agent Characteristics and Coordination**
>    Although the execution process is decentralized, true coordination is achieved through our training paradigm and the emergent behaviors of the learned policies. Algorithmically, SARL adopts the Centralized Training with Decentralized Execution (CTDE) paradigm. During training, the centralized Critic evaluates the joint state-action value function $Q(s_{joint}, a_{joint})$. This mechanism guides the Actor's policy gradients toward optimizing global airspace efficiency and safety, rather than merely pursuing selfish, greedy evasion. Furthermore, the integrated KSS module provides guidance based on aircraft states, training the model to learn correct collision avoidance maneuvers that comply with aviation rules. We will refine the description of this workflow in the revised manuscript. To verify action coordination, please refer to the visualization in Figure 6(b): in a close-proximity conflict, flights KL16 and KL23 executed upward and downward altitude adjustments, respectively, successfully and collaboratively resolving the conflict.
>
> 2. **Adversarial and Uncontrolled Scenarios**
>    Regarding your question about adversarial scenarios, this is indeed an excellent perspective on heterogeneous traffic systems, which is currently a very active area of exploration in low-altitude UAV domains. Our original experiments did not include this setup because our research primarily targets civil aviation, where all aircraft must strictly adhere to airspace management directives. However, to address your concern, we have conducted additional experiments for analysis. We introduced a certain number of non-cooperative aircraft (not controlled by the RL agents) into the environment. The results demonstrate that even against these uncontrolled aircraft, our PERG module retains the ability to capture their anomalous trajectories, enabling the agents to execute successful subsequent conflict resolutions.
> | Control Rate | Succ Rate (%) | Avg Op. Cost | Avg Min Sep |
> |--------------|---------------|--------------|-------------|
> | 100%         | 99.28         | 157.54       | 24.39       |
> | 66.6%        | 98.37         | 385.41       | 24.88       |
> | 50%          | 97.25         | 413.18       | 24.93       |
> | 33.3%        | 97.14         | 416.82       | 21.34       |
>
> 3. **Scenario Overfitting**
>    Regarding your concerns about potential scenario overfitting, we have supplemented the manuscript with additional experiments demonstrating the model's strong generalization capabilities. Furthermore, while testing across a wider variety of aircraft models certainly holds value, the fundamental physical constraints our network addresses (e.g., inertia, turn radius, and vertical speed limits) are conceptually consistent across all aircraft types.
> | Env          | Succ Rate (%) | Avg Op. Cost | Avg Min Sep |
> |--------------|---------------|--------------|-------------|
> | Train_Case   | 99.28         | 157.54       | 24.39       |
> | Eval_Case1   | 99.14         | 201.80       | 22.82       |
> | Eval_Case2   | 99.33         | 198.12       | 25.29       |
> | Eval_Case3   | 99.23         | 153.22       | 27.00       |
>
> 4. **Model Generalization Across Domains**
>    Although SARL is contextualized within the aviation domain, its core architecture is fundamentally a universal framework suitable for any high-density spatial navigation task involving continuous states and decoupled action constraints. The framework is highly transferable and can be broadly applied to autonomous vehicle platooning at complex intersections, swarm robot collaboration in warehousing and logistics, and maritime vessel collision avoidance. We believe our work not only provides a reproducible paradigm for intelligent air traffic navigation but also offers valuable insights for these adjacent fields.
>
> 5. **Computational Overhead**
>    We completely understand your concerns regarding the computational complexity typically associated with Graph Neural Networks. We wish to clarify that PERG does not construct a dense global graph with an $O(N^2)$ complexity. Instead, it constructs an ego-centric local graph, bounded by a specific physical detection radius. Therefore, for any given agent, the number of neighbors $K$ is practically limited by the physical capacity of that local airspace sector. Testing has shown that SARL's real-time inference latency is approximately 2.6 milliseconds; although this is higher than MADDPG's 1-millisecond latency, it remains more than adequate for practical deployment, considering that real-world ATC systems typically update data at a frequency measured in seconds.

---

### Official Review · Reviewer_FJpZ · 2026-03-23

**Soundness:** 3
**Presentation:** 2
**Significance:** 3
**Originality:** 3
**Overall Recommendation:** 3
**Confidence:** 4

**Summary:**

This paper studies multi-agent reinforcement learning for airspace conflict resolution. The main motivation is twofold. First, existing methods in aviation scenarios often suffer from a mismatch between perception and action design, while also facing a dynamically changing number of aircraft. Second, commonly used action formulations are continuously coupled, which does not align well with the sparse and decoupled nature of real-world air traffic control instructions.
To address these issues, the authors propose **SARL (Structure-Aligned Reinforcement Learning)**, a framework consisting of three main components:
1. **PERG**: a physics-enhanced relational graph based on graph attention, which encodes a dynamic number of neighboring aircraft into a graph representation and incorporates physical priors to improve state scalability.
2. **SC-MoE**: a sparse cognition-based mixture-of-experts action head, which uses a routing mechanism to select among three experts corresponding to speed, heading, and altitude, making the action design better aligned with practical air traffic control logic.
3. **KSS**: a rule-based kinematic safety shield, which corrects unsafe actions during inference and penalizes dangerous actions during training, thereby encouraging the learning of safer and more compliant behaviors.
Experiments are conducted in the BlueSky simulator. The proposed method is compared against several general MARL baselines as well as two aviation-oriented methods. Ablation studies are also provided to evaluate the contributions of PERG, SC-MoE, and KSS individually. Overall, the results suggest that SARL outperforms the compared methods on several metrics, including success rate, operational cost, and minimum separation.

**Compliance With Llm Reviewing Policy:**

Affirmed.

**Final Justification:**

The detailed response has alleviated my concerns, and I am therefore maintaining my score.

**Key Questions For Authors:**

1. The proposed method introduces explicit safety shielding and sparse action priors. Were the baseline methods given access to comparable rule-based priors, action constraints, and observation modeling improvements? Otherwise, it remains unclear whether the observed gains come primarily from system-level prior knowledge or from the learning framework itself. A stricter fairness-controlled comparison may be needed.
2. Regarding **KSS**, the paper mentions STL and aviation rules, but does not clearly specify the rule set, triggering conditions, conflict priorities, or the exact process of safety correction. Please provide a formal definition of KSS, along with illustrative examples, and explain how conflicts among multiple rules are handled.
3. Regarding the necessity of action sparsity: does the “single-command-per-timestep” design limit the ability to perform optimal avoidance in highly complex or emergency scenarios? Are there concrete cases showing that sparse actions are preferable to multi-dimensional continuous actions in this setting?

**Limitations:**

1. The experiments are conducted only in simulation, and there may be a significant sim-to-real gap when transferring to real-world conditions.
2. The KSS rule library may be incomplete, which could lead to over-conservative behavior or conflicts among rules.
3. The paper should discuss the risk of over-reliance on automation and the continued need for human oversight in safety-critical scenarios.

**Strengths And Weaknesses:**

##Strengths
1. The problem setting is clearly motivated and practically relevant. The paper identifies limitations of existing methods from the perspective of aligning perception and action structure, which is meaningful in this domain.
2. The overall framework is fairly comprehensive, combining a graph-based perception module (PERG), a sparse action-generation module (SC-MoE), and a safety shield (KSS). The design is coherent and appears well aligned with real-world operational requirements.
3. The empirical results in simulation are promising. SARL outperforms the compared methods on multiple evaluation metrics, suggesting potential practical value.
4. The paper includes ablation studies for all three proposed components, which provides initial support for their effectiveness.

##Weaknesses
1. The formalization of **KSS** is not sufficiently specific. Although the paper mentions STL and aviation rules, it does not provide a clear and detailed definition of the rule set, conflict handling mechanism, or implementation details. This weakens both the reproducibility and the credibility of the safety component.
2. While the overall framework is reasonable, the novelty of the individual technical components appears somewhat limited. The paper would benefit from a clearer discussion of how it substantively differs from prior work in graph-based reinforcement learning, mixture-of-experts architectures, and safe reinforcement learning.
3. The fairness of the experimental comparison requires further clarification. In particular, it is unclear whether the baselines were equipped with comparable observation modeling, action constraints, and safety mechanisms. As a result, it is difficult to disentangle performance gains due to stronger inductive biases and domain priors from gains due to the learning algorithm itself.

---

> ### Author Rebuttal · Authors · 2026-03-30
>
> We sincerely thank the reviewer for the constructive feedback and insightful comments. We address your specific concerns below:
>
> 1. **Fairness of Baseline Comparisons**
>    First, we would like to clarify that all baseline methods were evaluated under strictly fair conditions. They received the exact same kinematic observations and shared the identical reward design for the aircraft flight process. The incorporation of aviation rule-based constraints for feedback learning is precisely our core innovation compared to standard MARL in the aviation domain. To specifically address your concerns, we conducted an independent ablation analysis of these components within the "Ablation Experiments" section (as shown in Figure 5). Specifically, models incorporating only PERG and SC-MoE (without the KSS) still outperform standard baselines (e.g., MADDPG) in both Success Rate and Operational Cost. Further experiments demonstrate that this performance leap fundamentally stems from our neural network's structural optimizations—specifically, resolving dimensional mismatch and action coupling—rather than relying solely on the KSS as a safety net.
> | Method | Succ Rate (%) | Avg Op. Cost | Avg Min Sep |
> |--------|---------------|--------------|-------------|
> | MADDPG | 94.84 | 345.56 | 17.32 |
> | MADDPG w KSS | 97.11 | 525.52 | 19.65 |
> | MAPPO | 86.12 | 512.15 | 11.98 |
> | MAPPO w KSS | 93.50 | 945.07 | 17.27 |
> | **SARL** | **99.28** | **157.54** | **24.39** |
>
> 2. **Detailed Mechanism of the KSS**
>    We apologize for the omission of formal KSS definitions in the main text due to space constraints.
>    - **Trigger Conditions** The mechanism uses a constant velocity kinematic model to project the current state forward. A violation alert is triggered if the predicted distance at the Time to Closest Point of Approach is strictly less than the safety threshold and this event occurs within a predefined lookahead window.
>    - **Safety Correction & Priority** Upon triggering, the KSS overrides the RL agent's action with predefined evasive maneuvers. Following standard aviation heuristic rules, altitude adjustment is given the highest priority due to its rapid conflict resolution capability, followed by heading vector adjustments.
>    - **Multi-Rule Conflicts** In multi-agent scenarios with conflicting rules, the system adopts a decentralized "right-of-way" protocol (e.g., the aircraft on the right maintains its trajectory, while the yielding aircraft executes the KSS evasive maneuver).
>    We have compiled the complete formal Signal Temporal Logic (STL) definitions, trigger threshold settings, and pseudocode for the correction algorithms, which will be prominently featured in the revised manuscript.
>
> 3. **The Necessity of Action Sparsity**
>    In real-world civil aviation ATC, dispatchers strictly avoid issuing simultaneous, multi-dimensional commands (e.g., "descend 1000 feet, turn left 30 degrees, and reduce speed by 50 knots") to prevent pilot cognitive overload and operational errors. Our enforcement of action sparsity is explicitly designed to align with these physical operational standards. Restricting outputs to "one instruction per time step" does not preclude complex maneuvers; rather, these maneuvers are executed sequentially over the collision avoidance process (e.g., a right turn at time $t$, altitude adjustment at $t+1$, and returning to the original route at $t+2$). Therefore, substituting multi-dimensional continuous actions with sparse actions fundamentally aims to bridge the gap between RL outputs and actual aviation directives. This advantage is clearly illustrated in Figure 5(b): continuous multi-dimensional control yields an aggregate action rate exceeding 150%, requiring nearly 1.5 actions per time step. This overcomplicates ATC operations, representing a scenario where the agent chases high rewards while entirely ignoring practical feasibility.
>
> 4. **Simulation vs. Real-World Deployment**
>    The sim-to-real gap is indeed a critical consideration, but we have made every effort to minimize its impact. To mitigate this, our chosen simulation environment, BlueSky, is built strictly on real-world physical dynamics, capable of accurately modeling factors like wind speed and aircraft kinematic rules. Regarding aviation rules and human oversight, we are delighted that our perspectives align. We are actively planning to enhance human-machine interaction by addressing the interpretability of aviation decision-making and incorporating more human-in-the-loop directives. We greatly appreciate your validation and support for our future work in this direction.

---

> > ### Author Rebuttal · Reviewer_FJpZ · 2026-04-04
> >
> > Thank you for the detailed response which alleviates my concerns. I am therefore maintaining my score.

---

> > > ### Author Response · Authors · 2026-04-07
> > >
> > > Thank you for reviewing our rebuttal. We are very glad to hear that our detailed response successfully alleviated your concerns. We deeply appreciate the time, effort, and constructive feedback you have provided throughout this review process, which have been invaluable in improving the quality of our work.

---

### Decision · Program_Chairs · 2026-04-30

**Decision:**

Accept (regular)

**Comment:**

Aiming to bridge the gap between perception and action, this paper proposes SARL, combining the Physically Encoded Relational Graph (PERG) to solve the fixed input dimensionality problem, and the Sparse Cognitive Mixture-of-Experts (SC-MoE) to enhance decision stability. Some other technique as Kinematic Kafety Shield (KSS) is also introduced to improve inference-time safety. Empirical results in simulation are promising.

The authors' rebuttal was acknowledged partially by the reviewers, and the concerns on mechanism validation and scalability still remain.